# An Online Adaptive Sampling Algorithm for Stochastic Difference-of-convex Optimization with Time-varying Distributions

**Yuhan Ye** [1]  **Ying Cui** [2]  **Jingyi Wang** [3]

## Abstract

We propose an online adaptive sampling algorithm for solving stochastic nonsmooth difference-of-convex (DC) problems under time-varying distributions. At each iteration, the algorithm relies solely on data generated from the current distribution and employs distinct adaptive sampling rates for the convex and concave components of the DC function, a novel design guided by our theoretical analysis. We show that, under proper conditions on the convergence of distributions, the algorithm converges subsequentially to DC critical points almost surely. Furthermore, the sample size requirement of our proposed algorithm matches the results achieved in the smooth case or when a measurable subgradient selector is available, both under static distributions. A key element of this analysis is the derivation of a novel $O(\sqrt{p/n})$ pointwise convergence rate (modulo logarithmic factors) for the sample average approximation of subdifferential mappings, where $p$ is the dimension of the variable and $n$ is the sample size – a result of independent interest. Numerical experiments show that the algorithm is efficient for addressing online stochastic nonsmooth problems.

## 1. Introduction

We consider the class of stochastic nonsmooth nonconvex optimization problems in the form of

$$\underset{x \in C}{\text{minimize}} \; f(x) \triangleq \underbrace{\mathbb{E}_{\xi \sim P_\xi}[G(x, \xi)]}_{\triangleq g(x)} - \underbrace{\mathbb{E}_{\zeta \sim P_\zeta}[H(x, \zeta)]}_{\triangleq h(x)},$$

$$(1)$$

where $C \subset \mathbb{R}^p$ is a convex set, $\xi, \zeta \subset \Omega$ are random vectors with probability measures $P_\xi$, $P_\zeta$, respectively, and $G, H : (\mathbb{R}^p, \Omega) \to \mathbb{R}$ are Carathéodory functions, i.e., they are continuous in $x$ for all $\xi, \zeta \in \Omega$ and Borel measurable in $\xi$ and $\zeta$ for all $x \in C$. In addition, we assume $G$ and $H$ are convex in $x$ (though not necessarily smooth), making $f$ a difference-of-convex (DC) function.

When functions $g$ and $h$ are fully accessible, problem (1) can be solved via the classical DC algorithm (DCA). At each iteration, a convex subproblem is solved by linearizing $h$ via the subgradient at the previous point, *i.e.*, $x_{t+1} = \underset{x \in C}{\text{argmin}} \left[ g(x) - y_t^T(x - x_t) + \frac{\mu}{2}\|x - x_t\|^2 \right]$ for some $y_t \in \partial h(x_t)$ and $\mu > 0$. Due to the convexity of $h$, it can be shown that the objective sequence $\{f(x_t)\}$ is non-increasing, and the iterates asymptotically converge to a so-called DC critical point of problem (1).

However, in many applications, functions $g$ and $h$ are not fully known and can only be estimated from sampled data. This challenge is compounded when the underlying data-generating distribution is time-varying, as in the case of fluctuating demand. The convergence analysis of stochastic DCA is, therefore, significantly more complex than its deterministic counterpart, as it must account for the sample average approximation (SAA) error in both the convex component and the linearized concave component. The latter, in particular, is closely tied to the convergence rate of the SAA error for subdifferential mappings when $H$ is nonsmooth in $x$, introducing additional difficulty in the analysis.

In this paper, we propose an online adaptive sampling algorithm to solve problem (1). At each iteration, new data from the current distribution is used to construct a stochastic approximation of the linearized DC function, while previous samples are discarded. Unlike stochastic DCAs that aggregate past samples to compute current solutions, our method is more robust to distributional shifts occurring during the data generations along the iterations. The algorithm dynamically determines the sample sizes needed to estimate $g$ and $\partial h$, adapting to the optimization path throughout the process. Specifically, when the current iterate is far from critical points, less precise yet computationally inexpensive function values and subgradient estimates suffice. However,

---

[1]School of Mathematical Sciences, Peking University, Beijing, China [2]Department of Industrial Engineering and Operations Research, University of California, Berkeley, Berkeley, CA, United States [3]Center for Applied Scientific Computing, Lawrence Livermore National Laboratory, Livermore, CA, United States. Correspondence to: Yuhan Ye <yyh03@mit.edu>.

*Proceedings of the 42nd International Conference on Machine Learning*, Vancouver, Canada. PMLR 267, 2025. Copyright 2025 by the author(s).

as the iterates approach the critical points, higher accuracy in function and subgradient estimation becomes crucial for theoretical guarantees and effective practical performance.

We summarize the contribution of the paper as follows:

- We derive a novel $O(\sqrt{p/n})$ convergence rate (modulo logarithmic factors) for the expected pointwise SAA error of set-valued subdifferential mappings (Theorems 3.4 and 3.5), matching the convergence rate of single-valued gradient mappings in the smooth case, where $p$ is the dimension of the variable and $n$ is the sample size. Our results complement existing work (Davis and Drusvyatskiy, 2022; Ruan, 2024) on the uniform convergence rate for the SAA error for subdifferential mappings. We adopt a new proof technique that analyzes the one-sided deviation of subdifferential set-valued functions through their support functions.

- We propose an online adaptive stochastic framework for DC optimization under time-varying distributions. Unlike existing algorithms in the literature (Le Thi et al., 2024), which require a Borel measurable subgradient selector that is challenging to implement in practice, our algorithm allows the selection of any subgradient from the sampled subdifferential set. Furthermore, our algorithm operates under weak assumptions on the data generation process, allowing the underlying distributions to vary over time without necessarily matching the true distribution. We establish theoretical guarantees under the novel assumption that the cumulative Wasserstein-1 distance between successive distributions over iterations is bounded.

- Assume that we draw $N_{g,t}$ samples to estimate $g$ and $N_{h,t}$ samples to estimate $\partial h$ at time $t$. For any $\alpha_g \in (0, 1/2)$ and $\alpha_h \in (0, 1)$, we establish the almost sure convergence of the iterative sequence to a critical point under the condition that $\sum_{t \geq 0} \left( \frac{1}{N_{g,t}^{\alpha_g}} + \frac{1}{N_{h,t}^{\alpha_h}} \right) < \infty$. We further propose adaptive sampling strategy to adjust sample sizes at each step based on progress from the most recent iteration. In practice, the adaptive strategy enhances performance compared to its non-adaptive counterpart by potentially reducing the number of samples required during the initial stage of the algorithm.

## 1.1. Related Literature

**Non-asymptotic convergence analysis of SAAs.** An important step in our analysis is the error estimation of SAAs of $g$ and $\partial h$. The non-asymptotic convergence analysis of SAAs for expected functions has been well-studied in the existing literature; see, for example, the monograph Shapiro (2000). For the SAA convergence rate of subdifferentials, Xu (2010) demonstrates non-asymptotic, dimension-dependent high-probability bounds on the distance between the empirical and population subdifferentials under the Hausdorff metric. However, the population objective is essentially required to be smooth. In Mei et al. (2018), the authors discuss uniform convergence of gradients for smooth objectives under the assumption that the gradient is sub-Gaussian with respect to the population data. In Foster et al. (2018), the authors provide dimension-independent high-probability convergence rates of gradients for smooth Lipschitz generalized linear models, utilizing a "chain rule" for Rademacher complexity. These works do not directly examine the convergence behavior of subdifferential sets. More recently, Ruan (2024) achieves a tight $O(\sqrt{p/n})$ rate (modulo logarithmic factors) for the uniform convergence of weakly convex subdifferential mappings. This complements the $O(\sqrt[4]{p/n})$ uniform convergence rate of subdifferentials in Davis and Drusvyatskiy (2022). However, their result is based on the convex-smooth composite structure, as well as subexponential assumptions for random vector and process, see Assumption C in Ruan (2024).

**Stochastic and Online DC Optimization.** While deterministic DC algorithms have been extensively studied in existing literature (Le Thi and Pham Dinh, 2018), their stochastic counterparts have only recently gained attention (Thi et al., 2017; Le Thi et al., 2020). The first work that allowed both components in a DC problem to be nonsmooth was presented in Le Thi et al. (2022), where an SDCA scheme was proposed that stores all past information for constructing future subproblems. This approach achieves near-optimal sample size requirement by adding just one sample per DCA subproblem. Le Thi et al. (2024) pioneered the study of DCA in an online setting, eliminating the need to store historical information. Their approach resamples at each iteration and employs SAAs to approximate the linearized DC function using new samples, resulting in adaptive capabilities that offer a significant advantage over those in Le Thi et al. (2022). However, this method relies on the realization of a Borel measurable subgradient selector, as specified in Assumption 1 of Le Thi et al. (2024).

Moreover, non-asymptotic convergence of stochastic DC optimization has been studied in Nitanda and Suzuki (2017); Xu et al. (2019), which propose stochastic proximal DC algorithms by adding quadratic terms for DC subproblems. Nevertheless, these analyses rely on smoothness or Hölder continuity of the gradient, which are often too strong for many nonsmooth functions. Recent work in nonsmooth weakly convex optimization (Davis and Drusvyatskiy, 2018; Sun and Sun, 2022; Moudafi, 2022; Yao et al., 2022) has introduced Moreau envelope smoothing approximations for both components, enabling a non-asymptotic convergence analysis to nearly $\epsilon$-critical points for deterministic problems—a relaxed convergence criterion. These works have yet to establish complete non-asymptotic convergence for

non-smooth DC problems since a gap remains between nearly $\epsilon$-critical points and true critical points.

Recent studies have explored online optimization under distribution shifts, particularly within online convex optimization and stochastic approximation methods. Standard approaches typically assess performance through regret bounds relative to a defined measure of distribution shifts (e.g., Besbes et al. (2015); Fahrbach et al. (2023); Sankararaman and Narayanaswamy (2023)). Our proposed algorithm differs due to the nonsmooth nonconvex structure, where regret-based analysis is inapplicable, as our results rely on asymptotic convergence properties instead.

**Adaptive Sampling in Stochastic Optimization.** Adaptive sampling methods offer advantages over fixed-sample approaches, such as leveraging parallelism and generating iterates with reduced variance due to progressively increasing sample sizes. Adaptive strategies often use gradient approximation tests to regulate accuracy. Examples include norm-based tests (Carter, 1991; Byrd et al., 2012), inner product tests (Bollapragada et al., 2018), and other methods (Cartis and Scheinberg, 2018; Jin et al., 2021). For a comprehensive overview of adaptive sampling techniques, readers are referred to Curtis and Scheinberg (2020).

## 2. Preliminaries

We first summarize the notation used throughout the paper. We write $\mathbb{R}^p$ as the $p$-dimensional Euclidean space equipped with the inner product $\langle x, y \rangle = x^\top y$ and the induced norm $\|x\| \triangleq \sqrt{x^\top x}$. The symbol $\mathbb{B}(x, \delta)$ is used to denote the closed ball of radius $\delta > 0$ centered at a vector $x \in \mathbb{R}^p$. Let $A$ and $C$ be two nonempty subsets of $\mathbb{R}^p$. The distance from a vector $x \in \mathbb{R}^p$ to $A$ is defined as $\mathrm{dist}(x, A) \triangleq \inf_{y \in A} \|y - x\|$. The one-sided deviation of $A$ from $C$ is defined as $\mathbb{D}(A, C) \triangleq \sup_{x \in A} \mathrm{dist}(x, C)$. The Hausdorff distance between $A$ and $C$ is defined as $\mathbb{H}(A, C) := \max\{\mathbb{D}(A, C), \mathbb{D}(C, A)\}$.

We proceed by introducing fundamental concepts from nonsmooth analysis. For detailed discussions, we refer the reader to the monographs (Clarke, 1990; Rockafellar and Wets, 1998; Mordukhovich, 2006). Let $r : \mathcal{O} \to \mathbb{R}$ be a function defined on an open set $\mathcal{O} \subseteq \mathbb{R}^p$. The classical one-sided directional derivative of $r$ at $\bar{x} \in \mathcal{O}$ along the direction $d \in \mathbb{R}^p$ is defined as $r'(\bar{x}; d) \triangleq \lim_{t \downarrow 0} \frac{r(\bar{x} + td) - r(\bar{x})}{t}$. The function $r$ is said to be directionally differentiable at $\bar{x} \in \mathcal{O}$ if it is directionally differentiable along any direction $d \in \mathbb{R}^p$. In contrast, the Clarke directional derivative of $r$ at $\bar{x} \in \mathcal{O}$ along the direction $d \in \mathbb{R}^p$ is defined as $r^\circ(\bar{x}; d) \triangleq \limsup_{x \to \bar{x}, t \downarrow 0} \frac{r(x + td) - r(x)}{t}$, which is finite

when $r$ is Lipschitz continuous near $\bar{x}$.

The Clarke subdifferential of $r$ at $\bar{x}$ is the set $\partial_C r(\bar{x}) \triangleq \{v \in \mathbb{R}^p \mid r^\circ(\bar{x}; d) \geq v^\top d, \forall d \in \mathbb{R}^p\}$. If $r$ is strictly differentiable at $\bar{x}$, then $\partial_C r(\bar{x}) = \{\nabla r(\bar{x})\}$. We say that $r$ is Clarke regular at $\bar{x} \in \mathcal{O}$ if $r$ is directionally differentiable at $\bar{x}$ and $r^\circ(\bar{x}; d) = r'(\bar{x}; d)$ for all $d \in \mathbb{R}^p$. This Clarke regularity at $\bar{x}$ is equivalent to have $r(x) \geq r(\bar{x}) + \bar{v}^\top (x - \bar{x}) + o(\|x - \bar{x}\|)$ for any $\bar{v} \in \partial_C r(\bar{x})$, which is natural satisfied when $r$ is convex. Moreover, if a function fails to satisfy the Clarke regularity at $\bar{x}$, there does not exist an approximate linear lower bound of the original function based on the Clarke subdifferentials with a small $o$ error locally. Since the concept of Clarke subdifferential coincides with the usual subdifferential in convex analysis for a convex function, we simply refer to Clarke subgradient as subgradient in the remainder of the paper.

Let $\mathcal{A} : \mathbb{R}^p \rightrightarrows \mathbb{R}^m$ be a set-valued mapping. Its outer limit at $x \in \mathbb{R}^p$ is defined as $\limsup_{x \to \bar{x}} \mathcal{A}(x) := \bigcup_{x^\nu \to \bar{x}} \limsup_{\nu \to \infty} \mathcal{A}(x^\nu) = \{u \mid \exists\, x^\nu \to \bar{x}, \exists\, u^\nu \to u \text{ with } u^\nu \in \mathcal{A}(x^\nu)\}$. We say $\mathcal{A}$ is outer semicontinuous (osc) at $\bar{x} \in \mathbb{R}^p$ if $\limsup_{x \to \bar{x}} \mathcal{A}(x) \subseteq \mathcal{A}(\bar{x})$. Clarke subdifferential is outer semicontinuous, which is necessary in establishing subsequential convergence, by Proposition 6.6 in (Rockafellar and Wets, 1998). In addition, for a Lipschitz $r$, $\partial r(\bar{x})$ is locally bounded, see Theorem 9.13 in (Rockafellar and Wets, 1998).

A point $x^* \in \mathbb{R}^p$ is called *a DC critical* point if $0 \in \partial g(x^*) - \partial h(x^*)$, or equivalently $\partial g(x^*) \cap \partial h(x^*) \neq \emptyset$. In this paper, the terminology *critical point* refers to DC criticality, as defined in the literature on DC programming.

Next, we review some basics of random set-valued mappings and their expectations. Let $(\Omega, \mathcal{F}, P)$ be a probability space, and for fixed $x$, let $\mathcal{A}(x, \omega) : \Omega \to 2^{\mathbb{R}^p}$ be a general set-valued mapping taking values in closed subsets of $\mathbb{R}^p$. The expectation $\mathbb{E}[\mathcal{A}(x, \omega)]$ is defined as the set of $\mathbb{E}[A(x, \omega)]$ over all integrable selections, where integrability follows Aumann's sense (Aumann, 1965). It is well defined if $\mathbb{E}[\mathbb{H}(0, \mathcal{A}(x, \omega))] < \infty$. Let $r(x, \xi) : \mathbb{R}^p \times \Xi \to \mathbb{R}$ be a random lower semicontinuous function, where $\xi : (\Omega, \mathcal{F}, P) \to \Xi$ is a random vector with support $\Xi \subset \mathbb{R}^m$. If $r$ is $\kappa(\xi)$-Lipschitz in $x$, where $\mathbb{E}[\kappa(\xi)] < \infty$; and for any $x$, $r(x, \xi)$ is Clarke regular for a.e. $\xi$. Then, $\mathbb{E}[r(x, \xi)]$ is Clarke regular, and $\partial_x \mathbb{E}[r(x, \xi)] = \mathbb{E}[\partial_x r(x, \xi)]$, by Theorem 2.7.2 in (Clarke, 1990).

Throughout this paper, we assume that the sample space $\Omega$ is equipped with a metric $d(\cdot, \cdot)$, making it a metric space. Let $\mathbb{P}(\Omega)$ denote the set of Radon probability measures on $\Omega$, where each measure $P \in \mathbb{P}(\Omega)$ has a finite first moment. That is, $\mathbb{E}_{\xi \sim P}[d(\xi, \xi_0)] < \infty$ for some $\xi_0 \in \Omega$.

For $\mu, \nu \in \mathbb{P}(\Omega)$, their Wasserstein-1 distance is defined as

$$W_1(\mu, \nu) = \sup_{g \in \text{Lip}_1(\Omega)} \{\mathbb{E}_{X \sim \mu}[g(X)] - \mathbb{E}_{Y \sim \nu}[g(Y)]\},$$

where $\text{Lip}_1(\Omega)$ denotes the set of all Lipschitz functions $g : \Omega \to \mathbb{R}$ with the Lipschitz constant 1.

## 3. The Convergence Rate for the SAA Error of Subdifferential Mappings

In this section, we establish a novel pointwise convergence rate of $O(\sqrt{p/n})$ for subdifferential mappings, where $p$ is the dimension of the variable and $n$ is the sample size. This addresses a major challenge in subgradient-based stochastic nonsmooth problems: analyzing the sampling error of stochastic subgradients regarding the sample size.

For a random function $\varphi(\cdot, \omega) : \mathcal{D}_\varphi(\subseteq \mathbb{R}^p) \to \mathbb{R}$ and independent and identically distributed (i.i.d.) random variables $(\omega^1, \ldots, \omega^n) \triangleq \bar{\omega}^n$ drawn from the same distribution of $\omega$, we could use $\frac{1}{n} \sum_{k=1}^{n} \tau(x, \omega^k)$ as an SAA estimation of the subgradient of $\mathbb{E}_\omega[\varphi(x, \omega)]$, where $\tau(x, \omega^k)$ is a subgradient selector that satisfies $\tau(x, \omega^k) \in \partial_x \varphi(x, \omega^k)$.

In the smooth case, each $\tau(x, \omega^k)$ is an unbiased estimate of the expected gradient at $x$, since $\mathbb{E}_\omega[\nabla_x \varphi(x, \omega)] = \nabla \mathbb{E}_\omega[\varphi(x, \omega)]$. This leads to a straightforward $O\left(\frac{1}{n}\right)$ convergence rate for the squared error in relation to the sample size $n$, i.e.,

$$\mathbb{E}_{\bar{\omega}^n} \left| \frac{1}{n} \left(\sum_{k=1}^{n} \nabla_x \varphi(x, \omega^k)\right) - \nabla \mathbb{E}_\omega[\varphi(x, \omega)] \right|^2 \leq \frac{\sigma^2}{n}, \tag{2}$$

where $\sigma^2$ is the uniform variance of $\nabla_x \phi(x, \omega^k)$. However, this result does not directly extend to nonsmooth set-valued subdifferentials. Some studies impose an additional assumption that for any $x$, $\tau(x, \cdot)$ is Borel measurable with respect to $\omega$, enabling a similar convergence rate to (2). In practice, however, implementing a Borel measurable subgradient selector is challenging and often infeasible.

To address this challenge, we analyze the convergence rate of the sample average subdifferential mapping $\partial \bar{\varphi}(x) := \frac{1}{n} \sum_{k=1}^{n} \partial_x \varphi(x, \omega^k)$ to its expected counterpart $\partial \varphi(x) = \mathbb{E}_\omega[\partial_x \varphi(x, \omega)]$. We define the SAA error for $\partial \varphi(x, \cdot) : \Omega \to 2^{\mathbb{R}^p}$ as

$$\Delta_n(\varphi, x, \bar{\omega}^n) \triangleq \mathbb{H}\left(\frac{1}{n} \sum_{i=1}^{n} \partial_x \varphi(x, \omega^i), \mathbb{E}_\omega \partial_x \varphi(x, \omega)\right).$$

In the following, we shall develop a novel $O(\sqrt{p/n})$ convergence rate (modulo logarithmic factors) for $\Delta_n(\varphi, x, \bar{\omega}^n)$. Our results enable algorithms to select any subgradient from the sampled subdifferential set at each iteration while achieving a sampling error bound comparable to the smooth case.

We begin by introducing a lemma regarding the convergence rate of SAAs in expectation. This result is derived from the Rademacher average of the random function $\psi(x, \omega)$, as discussed in Corollary 3.2 of (Ermoliev and Norkin, 2013) and further explored in Theorem 10.1.5 of (Cui and Pang, 2021). Let $r$ be any positive scalar. For a random function $\psi(\cdot, \omega) : \mathcal{D}_\psi(\subseteq [0, r]^p) \to \mathbb{R}$ and i.i.d. random variables $(\omega^1, \ldots, \omega^n) = \bar{\omega}^n$ drawn from the distribution of $\omega$, we define the SAA error as $\delta_n(\psi, \bar{\omega}^n) := \sup_{x \in \mathcal{D}_\psi} \left| \frac{1}{n} \sum_{i=1}^{n} \psi(x, \omega^i) - \mathbb{E}_\omega \psi(x, \omega) \right|$. We then have the following basic estimates, see, e.g., Theorem 3.1 in (Ermoliev and Norkin, 2013).

**Lemma 3.1.** *(Basic Estimates). If functions $\psi(\cdot, \omega)$ are bounded by constant $M$ and Lipschitz continuous with constant $L_\psi$ in the first variable $x$ uniformly in $\omega$, then for any $\alpha \in (0, 1/2)$, $s > 0$, it holds that*

$$\mathbb{E}_{\bar{\omega}^n} \delta_n(\psi, \bar{\omega}^n) \leq 2\sqrt{p}\left(L_\psi r + \frac{M}{\sqrt{(1 - 2\alpha)e}}\right)/n^\alpha,$$

$$P\left\{\sqrt{n}\left|\delta_n(\psi, \bar{\omega}^n) - \mathbb{E}_{\bar{\omega}^n}\delta_n(\psi, \bar{\omega}^n)\right| \geq s\right\} \leq 2\exp\left\{-\frac{s^2}{2M^2}\right\}.$$

To analyze the asymptotic behavior of $\Delta_n(\varphi, x, \bar{\omega}^n)$, we need the following assumption.

**Assumption 3.2.** The function $\varphi(\cdot, \omega)$ is convex and Lipschitz continuous with Lipschitz constant $L_\varphi$, in terms of the first variable $x \in \mathcal{D}_\varphi$, uniformly in $\omega$.

The support function of a set $S$ is defined as $\sigma(u, S) \triangleq \sup_{s \in S} u^T s$. It is well known that $\sigma(u, S) = \sigma(u, \text{conv} S)$, where conv denotes the convex hull of $S$. Moreover, for any nonempty sets $S$ and $S'$, it follows from (Christian, 2002) that

$$\sigma(u, S + S') = \sigma(u, S) + \sigma(u, S'). \tag{3}$$

Furthermore, the Hömander's formula, according to Theorem II-18 in (Castaing and Valadier, 1977), states that for any two nonempty convex and compact subsets $A$ and $B$ of $\mathbb{R}^p$:

$$\mathbb{D}(A, B) = \max_{\|u\| \leq 1} (\sigma(u, A) - \sigma(u, B)). \tag{4}$$

Using the above formula, we derive the following lemma that converts our targeted quantity $\Delta_n(\varphi, x, \bar{\omega}^n)$ into the SAA error of support functions; see, e.g., (Xu, 2010). Its proof, as well as proofs for Theorems 3.4 and 3.5, can be found in the appendix.

**Lemma 3.3.** *Under Assumption 3.2, for any $x \in \mathcal{D}_\varphi$,*

$$\Delta_n(\varphi, x, \bar{\omega}^n) =$$

$$\max_{\|u\| \leq 1} \left| \frac{1}{n} \sum_{i=1}^{n} \sigma(u, \partial_x \varphi(x, \omega^i)) - \mathbb{E}_\omega[\sigma(u, \partial_x \varphi(x, \omega))] \right|.$$

We now derive the SAA convergence in expectation.

**Theorem 3.4.** *Under Assumption 3.2, for any $\alpha \in (0, 1/2)$,*

$$\sup_{x \in \mathcal{D}_\varphi} \mathbb{E}_{\bar{\omega}^n} \left[ \Delta_n \left( \varphi, x, \bar{\omega}^n \right) \right] \leq \frac{c}{n^\alpha},$$

*where $c \triangleq 2\sqrt{p}(2L_\varphi + L_\varphi/\sqrt{(1-2\alpha)e})$.*

*Moreover, for any $s > 0$,*

$$P \left\{ n^\alpha \Delta_n \left( \varphi, x, \bar{\omega}^n \right) \geq c + s \right\} \leq \exp \left\{ -s^2 / \left( 2L_\varphi^2 \right) \right\}.$$

**Remark 1.** The concentration-type probabilistic results in Lemma 3.1 and Theorem 3.4 are due to McDiarmid's bounded difference inequality. They will play an important role in the proof of Theorem 3.5.

Next, we strengthen the above theorem to bound the squared SAA error, which is the key result of this section.

**Theorem 3.5.** *Under Assumption 3.2, for any $\alpha \in (0, 1/2)$, $\alpha' \in (\alpha, 1/2)$, we have*

$$\sup_{x \in \mathcal{D}_\varphi} \mathbb{E}_{\bar{\omega}^n} \left[ \Delta_n \left( \varphi, x, \bar{\omega}^n \right)^2 \right] \leq \frac{c}{n^{2\alpha}},$$

*where $c \triangleq \hat{c} \left( \hat{c} + L_\varphi \frac{\sqrt{\alpha'}}{\sqrt{2(\alpha'-\alpha)e}} \right) + L_\varphi^2$ with $\hat{c} \triangleq \sqrt{p}(2L_\varphi + L_\varphi/\sqrt{(1-2\alpha')e})$.*

When $\varphi(\cdot, \omega)$ is smooth, Theorem 3.5 simply becomes

$$\sup_{x \in \mathcal{D}_\varphi} \mathbb{E}_{\bar{\omega}^n} \left[ \Delta_n \left( \varphi, x, \bar{\omega}^n \right)^2 \right] \leq \frac{L_\varphi^2}{n},$$

that is, $c = L_\varphi^2$ and $\alpha = 1/2$. This demonstrates that our result almost matches the SAA convergence rate in the smooth case. The tools we have developed here can play a crucial role in non-asymptotic convergence analysis of other (subgradient-based) stochastic nonsmooth problems. For example, it enables a "variance reduction" technique similar to that used in smooth optimization. (Bollapragada et al., 2018; Byrd et al., 2012)

# 4. The Algorithm and Convergence

Before presenting our algorithm, we first list all the needed assumptions for the stochastic functions $G$ and $H$.

**Assumption 4.1.** (Assumptions for Functions)

1. The feasible region $C$ is convex and closed, and there exists a scalar $\check{f}$ such that $f(x) > \check{f}$ for all $x \in C$.

2. $G(\cdot, \xi)$ is $\rho_g$-convex ($\rho_g \geq 0$) and $H(\cdot, \zeta)$ is $\rho_h$-convex ($\rho_h \geq 0$) over $C$ for almost every $\xi, \zeta \in \Omega$.

3. $G(\cdot, \xi)$ is $L_g$-Lipschitz continuous and $H(\cdot, \zeta)$ is $L_h$-Lipschitz continuous over $C$ for almost every $\xi, \zeta \in \Omega$.

4. For all $x \in C$, $G(x, \cdot)$ is $L_\xi$-Lipschitz continuous and $H(x, \cdot)$ is $L_\zeta$-Lipschitz continuous over $\Omega$.

## 4.1. The Algorithmic Framework

We assume that at time $t$, the data sets $S_{g,t} \triangleq \{\xi^{t,i}\}_{i=1}^{N_{g,t}}$ and $S_{h,t} \triangleq \{\zeta^{t,i}\}_{i=1}^{N_{h,t}}$ are generated from the distributions $P_{\xi,t}$ and $P_{\zeta,t}$, respectively, where the latter distributions may not be exactly the same as the true distributions $P_\xi$ and $P_\zeta$. Let $g_t(x) \triangleq \mathbb{E}_{\xi \sim P_{\xi,t}}[G(x, \xi)]$, $h_t(x) \triangleq \mathbb{E}_{\zeta \sim P_{\zeta,t}}[H(x, \zeta)]$, and $f_t(x) \triangleq g_t(x) - h_t(x)$. At time $t$ and iterate $x_t$, we use the data from $S_{g,t}$ to construct a stochastic estimate $\bar{g}_t(\cdot)$ of the function $g(\cdot)$, and the data from $S_{h,t}$ to construct a stochastic estimate $\bar{h}_t(x_t)$ of $h(x_t)$, as well as a stochastic estimate $\bar{y}_t$ of the subgradient $\partial h(x_t)$. The overall estimation model $\bar{M}_t(\cdot)$ is given by:

$$\bar{M}_t(d) \triangleq \bar{g}_t(x_t + d) - \bar{h}_t(x_t) - \bar{y}_t^T d + \frac{1}{2}\mu_t\|d\|^2, \quad (5)$$

where $\mu_t > 0$ is the proximal parameter. The convex subproblem to be solved at iteration $t$ is

$$\begin{aligned} \underset{d}{\text{minimize}} \quad & \bar{M}_t(d) \\ \text{subject to} \quad & x_t + d \in C. \end{aligned} \quad (6)$$

The first-order optimality condition of subproblem (6) at the unique optimal solution $\bar{d}_t$ is

$$\bar{z}_{t+1} - \bar{y}_t + \mu_t \bar{d}_t + \bar{v}_t = 0, \quad (7)$$

where $\bar{z}_{t+1} \in \partial \bar{g}_t(x_t + \bar{d}_t)$ and $\bar{v}_t \in \partial i_C(x_t + \bar{d}_t)$ with $i_C$ being the indicator function of $C$. Our proposed online stochastic proximal DC algorithm (ospDCA) framework is presented in Algorithm 1, while the exact rule to update the parameters $\mu_t, N_{g,t}, N_{h,t}$ will be discussed later.

---

**Algorithm 1** The ospDCA framework

---

1: Initialize $x_0, \mu_0, N_{g,0}, N_{h,0}$.
2: **for** $t = 0, 1, 2, \cdots$ **do**
3:     Generate i.i.d. samples $S_{g,t} = \{\xi^{t,i}\}_{i=1}^{N_{g,t}}$ and $S_{h,t} = \{\zeta^{t,i}\}_{i=1}^{N_{h,t}}$ from $P_{\xi,t}$ and $P_{\zeta,t}$, which are independent of the past samples.
4:     Construct the approximation model $\bar{M}_t(d)$ in (5) by setting $\bar{g}_t(x) = \frac{1}{N_{g,t}} \sum_{i=1}^{N_{g,t}} G\left(x, \xi^{t,i}\right)$, $\bar{h}_t(x) = \frac{1}{N_{h,t}} \sum_{i=1}^{N_{h,t}} H\left(x, \zeta^{t,i}\right)$, and select $\bar{y}_t \in \partial \bar{h}_t(x_t) = \frac{1}{N_{h,t}} \sum_{i=1}^{N_{h,t}} \partial_x H\left(x_t, \zeta^{t,i}\right)$.
5:     Solve the convex subproblem (6) to obtain $\bar{d}_t$.
6:     Set $x_{t+1} = x_t + \bar{d}_t$.
7:     Update $\mu_{t+1}, N_{g,t+1}, N_{h,t+1}$.
8: **end for**

---

Under Assumption 4.1, it is trivial to verify that $\bar{g}_t(x)$ and $g(x)$ are $L_g$-Lipschitz, $\rho_g$-convex; and $\bar{h}_t(x)$ and $h(x)$ are $L_h$-Lipschitz, $\rho_h$-convex.

Let $\mathcal{F}_t \triangleq \sigma\left(S_{g,1}, S_{h,1}, S_{g,2}, S_{h,2}, \ldots, S_{g,t-1}, S_{h,t-1}\right)$ be a filtration, i.e., an increasing sequence of $\sigma$-fields generated by the samples used in the past $t - 1$ iterations.

**Remark 2.** If there exists an isomorphic mapping $\phi$ from $(\Omega, \mathcal{F}_1, P_{\xi,t})$ to $(\Omega, \mathcal{F}_2, P_{\zeta,t})$, Step 3 of Algorithm 1 can be simplified when $N_{g,t} \geq N_{h,t}$, as follows:

1. Generate i.i.d. samples $S_{g,t} = \{\xi^{t,i}\}_{i=1}^{N_{g,t}}$ from the distribution of $\xi$, which are independent of previous samples.

2. For $i = 1, 2, \ldots, N_{h,t}$, set $\zeta^{t,i} = \phi(\xi^{t,i})$ and let $S_{h,t} = \{\zeta^{t,i}\}_{i=1}^{N_{h,t}}$.

A similar procedure applies when $N_{g,t} < N_{h,t}$.

### 4.2. Convergence Analysis

In this section, we present the convergence result of Algorithm 1 based on Assumptions 4.1. A brief outline of the convergence analysis is provided in the main text, with detailed proofs available in the appendix.

We first analyze the inexact sufficient descent property at the $t$-th iteration and derive the following inequality. The result and its proof is similar to the deterministic case, see, e.g., Theorem 3 in (Tao and An, 1997) and Theorem 3.7 in (Tao and An, 1998).

**Lemma 4.2.** *(The Sufficient Descent Property)* For any $y_t \in \partial h_t(x_t)$, the step $x_{t+1}$ from Algorithm 1 satisfies

$$f_t(x_t) - f_t(x_{t+1}) \geq (y_t - \bar{y}_t)^T \bar{d}_t + \left( \mu_t + \frac{\rho_g + \rho_h}{2} \right) \|\bar{d}_t\|^2$$
$$+ g_t(x_t) - \bar{g}_t(x_t) - g_t(x_{t+1}) + \bar{g}_t(x_{t+1}).$$

To further the analysis, the SAA error bound derived in Section 3 comes into play. By Lemma 3.1, we could derive the SAA error estimation for $g_t(x_t) - g_t(x_{t+1})$ as follows.

**Corollary 4.3.** *For any $\alpha_g \in (0, 1/2)$, we have*

$$\mathbb{E}\left[ \left| \bar{g}_t(x_{t+1}) - \bar{g}_t(x_t) - g_t(x_{t+1}) + g_t(x_t) \right| \Big| \mathcal{F}_t \right] \leq \frac{C_g}{\mu_t N_{g,t}^{\alpha_g}},$$

*where $C_g = 4\sqrt{p} L_g (L_g + L_h) \left( 2 + \frac{L_g}{\sqrt{(1-2\alpha_g)e}} \right)$.*

**Remark 3.** Note that we relax the assumption that $G(x, \xi)$ is globally uniformly bounded, as posed in Le Thi et al. (2024). Instead, we use the proximal term $\mu_t$ to ensure that $\bar{d}_t$ does not become too large. This guarantees that $G(x_t, \xi) - G(x_{t+1}, \xi)$ remains uniformly bounded with respect to $\mu_t$, which facilitates our SAA error analysis of $g_t(x_t) - g_t(x_{t+1})$ (see the proof of Corollary 4.3 for details).

The SAA error estimation for $\partial h_t(x_t)$ is a direct corollary of Theorem 3.5:

**Corollary 4.4.** *For any $\alpha_h \in (0, 1)$, $\alpha_h' \in (\alpha_h, 1)$, we have*

$$\sup_{x \in C} \mathbb{E}\left[ \mathbb{D}^2 \left( \partial \bar{h}_t(x_t), \partial h_t(x_t) \right) | \mathcal{F}_t \right] \leq \frac{C_h}{n^{\alpha_h}},$$

*where $C_h = \hat{C}_h \left( \hat{C}_h + L_h \frac{\sqrt{\alpha_h'}}{\sqrt{2(\alpha_h' - \alpha_h)e}} \right) + L_h^2$ with $\hat{C}_h = \sqrt{p}(2L_h + L_h/\sqrt{(1 - \alpha_h')e})$.*

**Remark 4.** With regard to the estimation error from sampling, Liu et al. (2022) assumes that the variance of the stochastic objectives is bounded. Similarly, Berahas et al. (2021) needs an unbiased gradient estimation with bounded variance in the study of stochastic sequential quadratic programming. Sequential quadratic programming is extended to the nonsmooth DC problems with smooth convex component in the deterministic Wang and Petra (2023) and stochastic settings Wang et al. (2023), where again a bounded variance of subgradient estimation is required. In Shashaani et al. (2018), the Monte Carlo estimate of the objective is also assumed to be unbiased, and its variance is uniformly bounded. The tools developed in Section 3 provide a tight SAA bound for $\partial h$, allowing us to derive a result analogous to the one in smooth optimization discussed above.

In the following lemma, we present the sufficient descent property in expectation.

**Lemma 4.5.** *At the $t$-th iteration, the following stands for any $c > 0$:*

$$\mathbb{E}\left[ f_t(x_t) - f_{t+1}(x_{t+1}) \mid \mathcal{F}_t \right]$$
$$\geq \left( \mu_t + \frac{\rho_g + \rho_h}{2} - c \right) \mathbb{E}\left[ \|\bar{d}_t\|^2 \mid \mathcal{F}_t \right] - \frac{C_g}{\mu_t N_{g,t}^{\alpha_g}}$$
$$- \frac{C_h}{4c N_{h,t}^{\alpha_h}} - L_\xi W_1(P_{\xi,t+1}, P_{\xi,t}) - L_\zeta W_1(P_{\zeta,t+1}, P_{\zeta,t}),$$
$$\tag{8}$$

*where $\alpha_g \in (0, 1/2)$ and $\alpha_h \in (0, 1)$ with corresponding constants $C_g$ and $C_h$ defined in Corollaries 4.3 and 4.4.*

The following analysis is conducted under the key assumptions stated below.

**Assumption 4.6.** (Assumptions for Distributions) The sequences $P_{\xi,t}$ and $P_{\zeta,t}$ converge to $P_\xi$ and $P_\zeta$ in Wasserstein-1 distance, that is,

$$\lim_{t \to \infty} W_1(P_{\xi,t}, P_\xi) = 0 \quad \text{and} \quad \lim_{t \to \infty} W_1(P_{\zeta,t}, P_\zeta) = 0,$$

Furthermore, the cumulative Wasserstein-1 distance between successive distributions, which measures the complexity of distribution shift on the data stream, is bounded:

$$\sum_{t=1}^{+\infty} W_1(P_{\xi,t}, P_{\xi,t-1}) < \infty, \text{ and } \sum_{t=1}^{+\infty} W_1(P_{\zeta,t}, P_{\zeta,t-1}) < \infty.$$

**Remark 5.** Since the Wasserstein-1 distance of some common distributions is easy to calculate or control, it is not hard to construct examples of time-varying exponential or uniform distributions that satisfy this assumption. A simple example is the regression problem with finite number of outliers or finite times of distribution shifts (due to the change

of environment). An example of online sparse robust regression will be provided in Section 6, where time-varying multivariate normal distributions satisfying the above assumption are considered.

**Assumption 4.7.** (Assumptions for Parameters)

(a) There exist $0 < \check{\mu} < \hat{\mu}$ such that $\check{\mu} \le \mu_t \le \hat{\mu}, \forall t \ge 0$.

(b) There exist $\alpha_g \in (0, 1/2), \alpha_h \in (0, 1)$ such that

$$\sum_{t \ge 0} \left( \frac{1}{N_{g,t}^{\alpha_g}} + \frac{1}{N_{h,t}^{\alpha_h}} \right) < \infty. \tag{9}$$

Now, we are ready to present the squared summable property of the iteration step $\{\bar{d}_t\}$, and its almost sure convergence to zero. These results are important for the later analysis.

**Theorem 4.8.** *Under Assumptions 4.6 and 4.7, we have*

$$\lim_{t \to \infty} \mathbb{E} \left[ \sum_{t \ge 0} \|\bar{d}_t\|^2 \, |\mathcal{F}_0 \right] < \infty, \text{ hence } \mathbb{E} \left[ \|\bar{d}_t\| \, |\mathcal{F}_0 \right] \to 0.$$

*Furthermore,* $\lim_{t \to \infty} \|\bar{d}_t\| = 0$ *with probability 1.*

To proceed, we first provide a technical Lemma, which concerns the law of large numbers (LLN) for SAA sequence.

**Lemma 4.9.** *Under Assumptions 4.6 and 4.7, for any fixed $R > 0, \hat{x} \in C, x \in \mathbb{B}(\bar{x}, R)$, the following limits hold as $t \to \infty$ with probability 1:*

$$\bar{g}_t(x) - \bar{g}_t(\hat{x}) - (g_t(x) - g_t(\hat{x})) \to 0,$$

$$\bar{h}_t(x) - \bar{h}_t(\hat{x}) - (h_t(x) - h_t(\hat{x})) \to 0.$$

We are ready to present our main convergence result, which is the best that can be achieved under stochastic nonconvex and nonsmooth conditions.

**Theorem 4.10.** *Under Assumptions 4.6 and 4.7, every accumulation point of the sequence $\{x_t\}$ produced by Algorithm 1 is a DC critical point of $f$ with probability 1.*

The above theorem only provides the asymptotic convergence of the algorithm, not the non-asymptotic complexity. The known complexity of the deterministic dc algorithm in (Le Thi et al., 2020) requires a smoothness assumption on either $g$ or $h$. We left it as a future work to derive the iteration complexity of our proposed algorithm.

The sample size requirement of our algorithm is presented in (9). Notably, the bounds on exponents $\alpha_g$ and $\alpha_h$ are different. To provide some intuition, this difference arises from the DC structure and the improved convergence rate of the SAA error for the subdifferential mapping. Specifically, linearizing the function $h$ couples the SAA error of $\partial h$ with

the stepsize $\bar{d}_t$, as demonstrated in Lemma 4.2. By applying the Cauchy-Schwarz inequality, we elevate the SAA error of $\partial h$ from first-order to second-order in expectation (see Lemma 4.5), for which Theorem 3.5 establishes the tight convergence rate.

According to Assumption 4.7 (a), the proximal terms $\mu_t$ for each DC subproblem can be pre-selected arbitrarily, as long as they are upper and lower bounded by positive constants. Regarding the sample size requirement given in Assumption 4.7 (b), this is inherent to our approach and difficult to avoid, as it ensures the necessary accuracy of the algorithm at each step. The choice of sample sizes and step sizes remains an active research topic in stochastic optimization. Even for stochastic gradient descent applied to smooth optimization problems, a non-diminishing step size selection requires sublinearly increasing sampling sizes to guarantee convergence.

## 5. An Adaptive Sampling Algorithm

In this section, we introduce an adaptive sampling strategy for updating $\mu_t, N_{g,t}, N_{h,t}$ in Algorithm 1. As discussed in the convergence analysis, the key requirement is to ensure that Assumption 4.7 holds. Since Assumption 4.7 (a) is relatively easy to satisfy, we mainly focus on developing strategies to satisfy Assumption 4.7 (b). Given pre-determined constants $c_l, c_\mu > 0$, a common approach is to increase the sample sizes sublinearly based on the following condition:

**Condition 5.1.** Suppose that $\hat{N}_{g,t}$ and $\hat{N}_{h,t}$ are pre-defined such that $\sum_{t \ge 0} \left( \hat{N}_{h,t}^{-\alpha_h} + \hat{N}_{g,t}^{-\alpha_g} \right) < \infty$, we say that the **Summable Condition** holds at the $t$-th iteration if the parameters $c_t, \mu_t, N_{g,t}, N_{h,t}$ are chosen to satisfy:

$$N_{g,t} \ge \hat{N}_{g,t}, N_{h,t} \ge \hat{N}_{h,t} \text{ and } c_l \le c_t \le \mu_t + \frac{\rho_g + \rho_h}{2} - c_\mu. \tag{10}$$

However, these pre-determined sample sizes do not adapt to the algorithm's progress at each iteration. In the following, we introduce a practical condition that determines $N_{g,t}$ and $N_{h,t}$ based on the optimization path. Intuitively, a larger stepsize in the early iterations suggests that the current point is far from critical points when less precise but computationally cheaper estimates are sufficient. In contrast, as the algorithm nears the critical points, the stepsize decreases, requiring more accurate estimations to ensure both theoretical guarantees and practical performance. Building on this intuition, we propose a practical **Stepsize Norm Condition** for adaptive sampling, where the sample size at each iteration is determined by the current stepsize.

**Condition 5.2.** We say that **Stepsize Norm Condition** stands at the $t$-th iteration if parameters $c_t, \mu_t, N_{g,t}, N_{h,t}$

are selected to satisfy:

$$\left(\mu_{t-1} - c_\mu - c_{t-1}\right) \|\bar{d}_{t-1}\|^2 \geq \frac{C_g}{\mu_t N_{g,t}^{\alpha_g}} + \frac{C_h}{4 c_t N_{h,t}^{\alpha_h}},$$

$$c_t \leq \mu_t + \frac{\rho_g + \rho_h}{2} - c_\mu. \tag{11}$$

**Remark 6.** Here, $c_t$ acts as an intermediate variable for parameter updates, linking others to ensure convergence. These variables serve only to determine $\mu_t$, $N_{g,t}$, and $N_{h,t}$.

As presented in the following theorem, Assumption 4.7 (b) stands when either condition is satisfied. This plays a critical role in designing a practical adaptive ospDCA with convergence guarantee. Compared to the gradient accuracy condition and other variance-based tests in (Byrd et al., 2012; Bollapragada et al., 2018), our adaptive sampling scheme is not only practically implementable but also backed by rigorous theoretical guarantees.

**Theorem 5.3.** *If either **Summable Condition** (10) or **Stepsize Norm Condition** (11) is satisfied for sufficiently large $t$, and Assumptions 4.7 (a) and 4.6 stand, then Assumption 4.7 (b) stands.*

To eliminate the intermediate variable $c_t$ and adapt the algorithm for any predetermined sequence $\{\mu_t\}$ satisfying $0 < \check{\mu} \leq \mu_t \leq \hat{\mu}$, we propose a simplified algorithm by fixing $c_t = \frac{\rho_g + \rho_h}{2} + \frac{\check{\mu}}{4} = c_l$ and setting $c_\mu = \frac{\check{\mu}}{4}$, as detailed in Algorithm 2. The complete version of the adaptive sampling ospDCA can be found in the appendix; see Algorithm 3.

**Remark 7.** Consider the subproblem when updating the sample size $N_{g,t}$ and $N_{h,t}$. In order to minimize the total number of samples, one could derive that $N_{h,t} = \sqrt{\frac{2 C_h \mu_{t+1}}{C_g (2\rho_g + 2\rho_h + \check{\mu})}} N_{g,t}^{3/4}$. Hence the optimal order of $N_{h,t}$ is $O(N_{g,t}^{3/4})$. Furthermore, if the updating rule of $N_{g,t}$ and $N_{h,t}$ is based on this result, then sample size upper bound sequence $\hat{N}_{h,t}$ is no longer required.

# 6. An Application: Online Sparse Robust Regression

We consider the online linear regression problem with a robust loss and sparsity-promoting DC regularization. Given streaming data $\{(x_i, y_i)\}_{i=1}^{\infty}$ drawn from unknown and varying distributions $\mathcal{D}_t$, the optimization problem is formulated as minimizing the expected objective:

$$\min_{\beta \in \mathbb{R}^p} \mathbb{E}_{(x,y) \sim \mathcal{D}_t} \left[ |y - \langle \beta, x \rangle| \right] + \lambda \sum_{j=1}^{p} \min(1, \alpha |\beta_j|).$$

The regularization term $\sum_{j=1}^{p} \min(1, \alpha |\beta_j|)$ is a capped-$\ell_1$ penalty, which approximates the sparsity-inducing $\ell_0$-norm. To facilitate optimization, we use the following DC

**Algorithm 2** Adaptive ospDCA

**Require:** Initial point $x_0$, error estimation parameter $\alpha_g \in (0, 1/2)$, $\alpha_h \in (0, 1)$ with corresponding $C_g, C_h$ defined in Corollaries 4.3 and 4.4, sample size upper bound sequence $\{\hat{N}_{g,t}\}$ and $\{\hat{N}_{h,t}\}$ which satisfy $\sum_{t \geq 0} \left( \hat{N}_{h,t}^{-\alpha_h} + \hat{N}_{g,t}^{-\alpha_g} \right) < \infty$, predetermined proximal parameters $\{\mu_t\}$ with upper bound $\hat{\mu}$ and lower bound $\check{\mu}$.

1: **for** $t = 0, 1, 2, \cdots$ **do**

2:     Generate i.i.d. samples $\{\xi^{t,i}\}_{i=1}^{N_{g,t}}$ and $\{\zeta^{t,i}\}_{i=1}^{N_{h,t}}$ from the distribution of $\xi$ and $\zeta$, which are independent of the past samples.

3:     Set $\bar{g}_t(x) = \frac{1}{N_{g,t}} \sum_{i=1}^{N_{g,t}} G(x, \xi^{t,i})$, $\bar{h}_t(x) = \frac{1}{N_{h,t}} \sum_{i=1}^{N_{h,t}} H(x, \zeta^{t,i})$, and select $\bar{y}_t \in \partial \bar{h}_t(x_t)$.

4:     Solve the convex subproblem to obtain $\bar{d}_t$:

$$\begin{aligned} \underset{d}{\text{minimize}} \quad & \bar{g}_t(x_t + d) - \bar{h}_t(x_t) - \bar{y}_t^T d + \frac{1}{2} \mu_t \|d\|^2 \\ \text{subject to} \quad & x_t + d \in C. \end{aligned}$$

5:     Set $x_{t+1} = x_t + \bar{d}_t$.

6:     Update $N_{g,t+1}$ and $N_{h,t+1}$ such that one of the followings stands:

      1. $\left(\mu_t - \frac{\check{\mu}}{2}\right) \|\bar{d}_t\|^2 \geq \frac{C_g}{\mu_{t+1} N_{g,t+1}^{\alpha_g}} + \frac{C_h}{(2\rho_g + 2\rho_h + \check{\mu}) N_{h,t+1}^{\alpha_h}}$,

      2. $N_{g,t+1} \geq \hat{N}_{g,t+1}$, and $N_{h,t} \geq \hat{N}_{h,t+1}$.

7: **end for**

decomposition: $\min(1, \alpha |\beta_j|) = 1 + \alpha |\beta_j| - \max(1, \alpha |\beta_j|)$. Thus, the final problem formulation in expectation form is:

$$\min_{\beta \in \mathbb{R}^p} \mathbb{E}_{(x,y) \sim \mathcal{D}_t} \left[ G(\beta, x, y) \right] - h(\beta),$$

where $G(\beta, x, y) = |y - \langle \beta, x \rangle| + \lambda \sum_{j=1}^{p} \left(1 + \alpha |\beta_j|\right)$, $h(\beta) = \sum_{j=1}^{p} \max(1, \alpha |\beta_j|)$. This expectation-based formulation enables efficient online optimization, making it well-suited for large-scale and streaming data scenarios.

**Baselines.** We implemented four baselines to compare with our proposed adaptive ospDCA. The first one is ospDCA with a pre-determined, sublinearly growing sample size of $t^{2.1}$ per iteration, without adaptivity. The second baseline is S(p)DCA, introduced in Le Thi et al. (2024), where we added an additional proximal term. This algorithm draws one new sample per iteration and uses aggregated samples to construct sample averages. The third and fourth baselines are ospDCA with a fixed sample size per iteration, using 100 and 1000 new samples for SAA, respectively.

**Datasets and Setup.** For the problem, we set $\alpha = 1$, $\lambda = 0.01$, and generate synthetic datasets. Specifically, at each time step $t$, the feature vector $x_t$ is sampled uniformly from

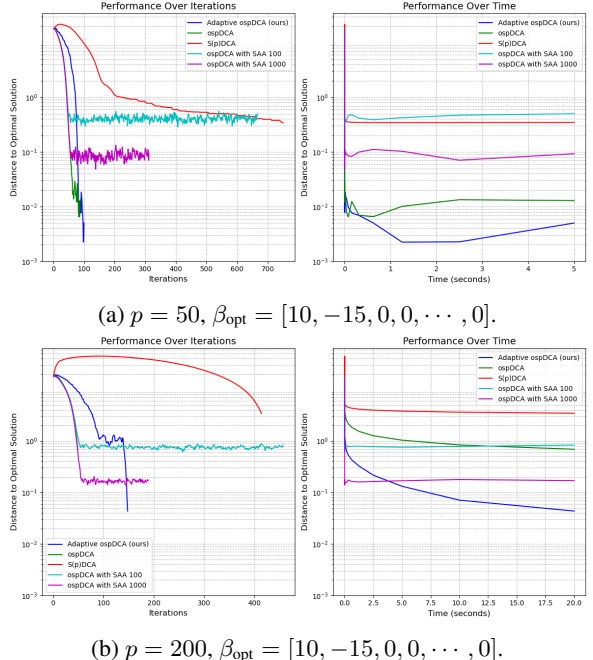

(a) $p = 50$, $\beta_{\text{opt}} = [10, -15, 0, 0, \cdots, 0]$.

(b) $p = 200$, $\beta_{\text{opt}} = [10, -15, 0, 0, \cdots, 0]$.

*Figure 1.* Algorithm behavior for online sparse robust regression.

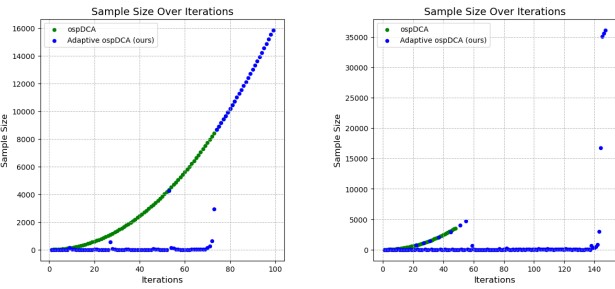

Sample size for experiment (a).    Sample size for experiment (b).

*Figure 2.* Sample size per iteration.

$[-1, 1]^p$. The corresponding label is given by

$$y_t = x_t^\top (\beta_{\text{opt}} + \delta_t) + \varepsilon,$$

where $\beta_{\text{opt}}$ is a known sparse optimal solution with nonzero entries at specific locations, $\varepsilon \sim N(0, 1)$ represents additive noise, and $\delta_t$ denotes a time-dependent distribution shift. It follows that $W_1(\mathcal{D}_t, \mathcal{D}_{t+1}) \leq \|\delta_t - \delta_{t+1}\|_1$ . In order to ensure that the cumulative Wasserstein-1 distance for $\mathcal{D}_t$ remains bounded, we set $\delta_t = (-1)^t 100 t^{-2} \mathbf{1}_p$, where $\mathbf{1}_p$ represents a $p$-dimensional column vector where all entries are equal to 1. We initialize $\beta$ at zero, set the proximal coefficient $\mu_t = 1$, $\alpha_g = 0.45 \in (0, 1/2)$, and run the experiment until a predefined runtime limit is reached. It is straightforward to verify that $G(\cdot, x, y)$ is 1-Lipschitz for every $x, y$, and $h(\cdot)$ is $\lambda\alpha$-Lipschitz. Furthermore, if we impose a bounded constraint on $\beta$, then $G(\beta, \cdot, \cdot)$ is also uniformly Lipschitz in $(x, y)$ for every $\beta$.

**Results.** We evaluate the performance by tracking the distance between the current iterate $\beta_t$ and the optimal solution $\beta_{\text{opt}}$. We plot the evolution of convergence error and computational time in Figures 1 and 3. Across all experiments, the performance of adaptive ospDCA consistently surpasses the baseline methods. This demonstrates that our proposed algorithm significantly improves convergence efficiency.

During early iterations, the sample size of adaptive ospDCA is relatively small, leading to reduced precision but higher computational efficiency. As the iteration points approach the optimal solution, the sample size increases to enhance estimation accuracy. This transition leads to faster progress in later iterations, ultimately surpassing other algorithms. Compared to its non-adaptive counterpart, adaptive ospDCA invests more time and samples in the later iterations (which are closer to the optimal and thus more important), as illustrated in Figures 2 and 4. The adaptivity makes it more efficient overall and more robust to distribution shifts. Additional experimental results are provided in the appendix.

## 7. Conclusion

In this work, we propose an efficient online adaptive sampling algorithm for stochastic nonsmooth difference-of-convex (DC) optimization problems with time-varying data distributions. The algorithm relies only on samples drawn from the distribution at the current iterate and adopts distinct adaptive sampling rates for the convex and concave components of the DC objective. We further prove that, under mild convergence conditions on the non-stationary distributions, the generated sequence almost surely has a subsequence that converges to a DC critical point. One of the core contribution of this paper lies in generalizing previous results in the field of stochastic online DC optimization to a broader class of nonsmooth DC problems with time-varying distributions, while maintaining a sampling size requirement comparable to the smooth case. Numerical experiments demonstrate that our algorithm performs well on online sparse robust regression tasks.

## Acknowledgements

Yuhan Ye is partially supported by the Elite Undergraduate Training Program of the School of Mathematical Sciences at Peking University. Ying Cui is supported by the National Science Foundation under Grants CCF-2416172 and DMS-2416250, and the National Institutes of Health under Grant 1R01CA287413-01. Jingyi Wang's work is performed under the auspices of the U.S. Department of Energy by Lawrence Livermore National Laboratory under Contract DE-AC52-07NA27344. Release number LLNL-JRNL-846514.

## Impact Statement

This paper presents work whose goal is to advance the field of Optimization. There are many potential societal consequences of our work, none of which we feel must be specifically highlighted here.

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

# A. Convergence Rate for the SAA Error of Subdifferential Mappings

## A.1. Proof of Lemma 3.3

*Proof.* By direction computation, we have

$$
\mathbb{D}\left(\frac{1}{n}\sum_{i=1}^{n}\partial_x\varphi\left(x,\omega^t\right), \mathbb{E}_\omega\left[\partial_x\varphi(x,\omega)\right]\right)
$$

$$
= \mathbb{D}\left(\operatorname{conv}\left\{\frac{1}{n}\sum_{i=1}^{n}\partial_x\varphi\left(x,\omega^i\right)\right\}, \operatorname{conv}\mathbb{E}_\omega\left[\partial_x\varphi(x,\omega)\right]\right)
$$

$$
= \sup_{\|u\|\leqslant 1}\left[\sigma\left(u,\frac{1}{n}\sum_{i=1}^{n}\partial_x\varphi\left(x,\omega^i\right)\right) - \sigma\left(u,\mathbb{E}_\omega\left[\partial_x\varphi(x,\omega)\right]\right)\right]
$$

$$
= \sup_{\|u\|\leqslant 1}\left[\frac{1}{n}\sum_{i=1}^{n}\sigma\left(u,\partial_x\varphi\left(x,\omega^i\right)\right) - \sigma\left(u,\mathbb{E}_\omega\left[\partial_x\varphi(x,\omega)\right]\right)\right]
$$

$$
= \sup_{\|u\|\leqslant 1}\left[\frac{1}{n}\sum_{i=1}^{n}\sigma\left(u,\partial_x\varphi\left(x,\omega^i\right)\right) - \mathbb{E}_\omega\left[\sigma\left(u,\partial_x\varphi(x,\omega)\right)\right]\right].
$$

By the convexity of $\varphi\left(x,\omega\right)$, $\partial_x\varphi\left(x,\omega^t\right)$ is convex and compact, hence $\frac{1}{n}\sum_{i=1}^{n}\partial_x\varphi\left(x,\omega^i\right)$ and $\mathbb{E}_\omega\left[\partial_x\varphi(x,\omega)\right]$ are convex. The first equality stands. The second equality is due to (4). The third equality is due to (3). The last equality is due to the interchangeability of $\mathbb{E}_\omega$ and $\sigma$; see Proposition 3.4 in (Papageorgiou, 1985) for details.

Following the same derivation, we also have

$$
\mathbb{D}\left(\mathbb{E}_\omega\left[\partial_x\varphi(x,\omega)\right], \frac{1}{n}\sum_{i=1}^{n}\partial_x\varphi\left(x,\omega^i\right)\right) = \sup_{\|u\|\leqslant 1}\left[\mathbb{E}_\omega\left[\sigma\left(u,\partial_x\varphi(x,\omega)\right)\right] - \frac{1}{n}\sum_{i=1}^{n}\sigma\left(u,\partial_x\varphi\left(x,\omega^i\right)\right)\right].
$$

We thus conclude the proof using the definition of the Hausdorff distance $\mathbb{H}$. $\square$

## A.2. Proof of Theorem 3.4

*Proof.* For any $x\in\mathcal{D}_\varphi$, by Lemma 3.3,

$$
\mathbb{E}_{\bar{\omega}^n}\left[\Delta_n\left(\varphi,x,\bar{\omega}^n\right)\right] = \mathbb{E}_{\bar{\omega}^n}\sup_{\|u\|\leqslant 1}\left|\frac{1}{n}\sum_{i=1}^{n}\psi(u,\omega^i) - \mathbb{E}_\omega\left[\psi(u,\omega)\right]\right|,
$$

where $\psi(u,\omega) = \sigma\left(u,\partial_x\varphi\left(x,\omega\right)\right)$. To satisfy the condition in Lemma 3.1, we first verify that $\psi(\cdot,\omega)$ are uniformly bounded by constant $L_\varphi$ and Lipschitz continuous with constant $L_\varphi$ in the first variable $u\in\mathbb{B}(0,1)\subseteq[-1,1]^p$ uniformly in $\omega$.

The first property is trivial since $\sup_{s\in\partial_x\varphi(x,\omega)}\|s\|\leq L_\varphi$. Now, we prove the second property. For any $u,v\in\mathbb{B}(0,1)$, suppose that $\sigma\left(u,\partial_x\varphi\left(x,\omega\right)\right) = u^T s$ where $s\in\partial_x\varphi\left(x,\omega\right)$. Then we have

$$
\sigma\left(v,\partial_x\varphi\left(x,\omega\right)\right) \geq v^T s \geq u^T s - \|u-v\|\|s\| \geq \sigma\left(u,\partial_x\varphi\left(x,\omega\right)\right) - L_\varphi\|u-v\|.
$$

Similarly, $\sigma\left(u,\partial_x\varphi\left(x,\omega\right)\right) \geq \sigma\left(v,\partial_x\varphi\left(x,\omega\right)\right) - L_\varphi\|u-v\|$, hence $\psi(u,\omega)$ is $L_\varphi$-Lipschitz continuous in $\mathbb{B}(0,1)$.

We thus finish the proof after using Lemma 3.1. $\square$

## A.3. Proof of Theorem 3.5

*Proof.* For any $x\in\mathcal{D}_\varphi$, let $\delta = \Delta_n\left(\varphi,x,\bar{\omega}^n\right)$, which is bounded by $L_\varphi$. By Lemma 3.4, $\mathbb{E}_{\bar{\omega}^n}[\delta] \leq \frac{\hat{c}}{n^\alpha}$ and $P\left\{\delta \geq \frac{\hat{c}+s}{n^\alpha}\right\} \leq \exp\left\{-s^2/\left(2L_\varphi^2\right)\right\}$ stands for any $s>0$, $\alpha'\in(\alpha,1/2)$.

Hence for any $s > 0$, we have

$$\mathbb{E}_{\bar{\omega}^n}[\delta^2] = \mathbb{E}_{\bar{\omega}^n}\left[\delta^2 \mathbf{1}(\delta < \frac{\hat{c}+s}{n^{\alpha'}})\right] + \mathbb{E}_{\bar{\omega}^n}\left[\delta^2 \mathbf{1}(\delta \geq \frac{\hat{c}+s}{n^{\alpha'}})\right]$$

$$\leq \mathbb{E}_{\bar{\omega}^n}[\delta]\frac{\hat{c}+s}{n^{\alpha'}} + L_\varphi^2 P\left\{\delta \geq \frac{\hat{c}+s}{n^{\alpha'}}\right\}$$

$$\leq \frac{\hat{c}(\hat{c}+s)}{n^{2\alpha'}} + L_\varphi^2 \exp\left\{-s^2/\left(2L_\varphi^2\right)\right\},$$

where the first inequality is due to $\delta \leq L_\varphi$, and the last inequality is due to Lemma 3.4.

Take $s = \sqrt{2}L_\varphi\sqrt{\alpha' \ln n}$, we could obtain that

$$\mathbb{E}_{\bar{\omega}^n}[\delta^2] \leq \frac{\hat{c}\left(\hat{c} + \sqrt{2}L_\varphi\sqrt{\alpha' \ln n}\right) + L_\varphi^2}{n^{2\alpha'}}.$$

Since $\sqrt{\ln n} \leq \frac{n^{2\alpha'-2\alpha}}{2\sqrt{(\alpha'-\alpha)e}}$, we have for any $\alpha' \in (\alpha, 1/2)$,

$$\mathbb{E}_{\bar{\omega}^n}[\delta^2] \leq \frac{\hat{c}\left(\hat{c} + \sqrt{2}L_\varphi\frac{\sqrt{\alpha'}}{2\sqrt{(\alpha'-\alpha)e}}\right) + L_\varphi^2}{n^{2\alpha}},$$

where $\hat{c} = \sqrt{p}(2L_\varphi + L_\varphi/\sqrt{(1-2\alpha')e})$. $\qquad\square$

## B. Convergence Analysis

### B.1. Proof of Lemma 4.2

*Proof.* From the convexity of $\bar{g}_t(\cdot)$ and $h_t(\cdot)$, we have

$$\bar{g}_t(x_t) - \bar{g}_t(x_t + \bar{d}_t) + \bar{z}_{t+1}^T \bar{d}_t \geq \frac{1}{2}\rho_g\|\bar{d}_t\|^2,$$

$$h_t(x_t + \bar{d}_t) - h_t(x_t) - y_t^T \bar{d}_t \geq \frac{1}{2}\rho_h\|\bar{d}_t\|^2,$$

$\qquad(12)$

for $\bar{z}_{t+1} \in \bar{\partial}\bar{g}_t(x_t + \bar{d}_t)$, $x_t + \bar{d}_t \in C$. Therefore,

$$f_t(x_t) - f_t(x_{t+1}) = g_t(x_t) - h_t(x_t) - g_t(x_{t+1}) + h_t(x_{t+1})$$
$$= \bar{g}_t(x_t) - \bar{g}_t(x_{t+1}) - h_t(x_t) + h_t(x_{t+1}) + [g_t(x_t) - \bar{g}_t(x_t)] - [g_t(x_{t+1}) - \bar{g}_t(x_{t+1})]$$
$$\geq - \bar{z}_{t+1}^T \bar{d}_t + y_t^T \bar{d}_t + \frac{1}{2}(\rho_g + \rho_h)\|\bar{d}_t\|^2 + g_t(x_t) - \bar{g}_t(x_t) - g_t(x_{t+1}) + \bar{g}_t(x_{t+1}).$$

$\qquad(13)$

Taking the dot product with $-\bar{d}_t$ of the first line of (7), we have

$$-\bar{z}_{t+1}^T \bar{d}_t + \bar{y}_t^T \bar{d}_t = \mu_t\|\bar{d}_t\|^2 + \bar{v}_t \bar{d}_t$$
$$\geq \mu_t\|\bar{d}_t\|^2 + [i_C(x_t) - i_C(x_t + \bar{d}_t) - \bar{v}_t^T(-\bar{d}_t)]$$
$$\geq \mu_t\|\bar{d}_t\|^2,$$

$\qquad(14)$

where the first inequality uses the second line of (7) and the second inequality comes from the convexity of $C$ and $i_C(\cdot)$. Applying (14) to (13) leads to

$$f_t(x_t) - f_t(x_{t+1}) \geq (y_t - \bar{y}_t)^T \bar{d}_t + \left(\mu_t + \frac{\rho_g + \rho_h}{2}\right)\|\bar{d}_t\|^2 + g_t(x_t) - \bar{g}_t(x_t) - g_t(x_{t+1}) + \bar{g}_t(x_{t+1}). \qquad(15)$$

$\qquad\square$

## B.2. Proof of Corollary 4.3

*Proof.* First we prove that $\|\bar{d}_t\| \leq \frac{2(L_g+L_h)}{\mu_t}$. Since $\bar{d}_t$ is the solution of subproblem (6), we have

$$\bar{g}_t(x_t + \bar{d}_t) - \bar{h}_t(x_t) - \bar{y}_t^T \bar{d}_t + \frac{1}{2}\mu_t\|\bar{d}_t\|^2 = \bar{M}_t(\bar{d}_t) \leq \bar{M}_t(0) = \bar{g}_t(x_t) - \bar{h}_t(x_t).$$

Hence we have $\frac{1}{2}\mu_t\|\bar{d}_t\|^2 \leq \bar{g}_t(x_t) - \bar{g}_t(x_t + \bar{d}_t) + \bar{y}_t^T \bar{d}_t \leq \|\bar{d}_t\|(L_g + L_h)$, which implies $\|\bar{d}_t\| \leq \frac{2(L_g+L_h)}{\mu_t}$.

Let $r_t = \frac{2(L_g+L_h)}{\mu_t}$ and $\psi(x, \xi) = G(x, \xi) - G(x_t, \xi)$, which is $L_g$-Lipschitz. By Lemma 3.1,

$$\mathbb{E}_{S_{g,t}}\left[\sup_{\delta_x \in [-r_t, r_t]^d} \left|\frac{1}{n}\sum_{i=1}^n \psi\left(x_t + \delta_x, \xi^{t,i}\right) - \mathbb{E}_{\xi \sim P_{\xi,t}}\psi(x_t + \delta_x, \xi)\right| \mid \mathcal{F}_t\right] \leq \frac{C_g}{N_{g,t}^\alpha},$$

where $C_g = 2\sqrt{p}(2L_g r_t + L_g r_t/\sqrt{(1-2\alpha)e})$. Noticed that

$$|\bar{g}_t(x_{t+1}) - \bar{g}_t(x_t) - g_t(x_{t+1}) + g_t(x_t)| \leq \sup_{\delta_x \in [-r_t, r_t]^d} \left|\frac{1}{n}\sum_{i=1}^n \psi\left(x_t + \delta_x, \xi^{t,i}\right) - \mathbb{E}_{\xi \sim P_{\xi,t}}\psi(x_t + \delta_x, \xi)\right|,$$

Hence, we finish the proof by substituting $r_t$ for its definition. $\qquad\square$

## B.3. Proof of Lemma 4.5

*Proof.* By Lemma 4.2, for any $y_t \in \partial h_t(x_t)$, $c > 0$,

$$f_t(x_t) - f_t(x_{t+1}) \geq (y_t - \bar{y}_t)^T \bar{d}_t + \mu_t + \frac{\rho_g + \rho_h}{2}\left\|\bar{d}_t\right\|^2 + g_t(x_t) - \bar{g}_t(x_t) - g_t(x_{t+1}) + \bar{g}_t(x_{t+1})$$

$$\geq \left(\mu_t + \frac{\rho_g + \rho_h}{2} - c\right)\left\|\bar{d}_t\right\|^2 + g_t(x_t) - \bar{g}_t(x_t) - g_t(x_{t+1}) + \bar{g}_t(x_{t+1}) - \frac{1}{4c}\left\|y_t - \bar{y}_t\right\|^2.$$

Take $y_t$ such that $dist(\bar{y}_t, \partial h_t(x_t)) = \|y_t - \bar{y}_t\|$, we have

$$f_t(x_t) - f_t(x_{t+1}) \geq \left(\mu_t + \frac{\rho_g + \rho_h}{2} - c\right)\left\|\bar{d}_t\right\|^2 + g_t(x_t) - \bar{g}_t(x_t) - g_t(x_{t+1}) + \bar{g}_t(x_{t+1}) - \frac{1}{4c}dist(\bar{y}_t, \partial h_t(x_t))^2$$

$$\geq \left(\mu_t + \frac{\rho_g + \rho_h}{2} - c\right)\left\|\bar{d}_t\right\|^2 + g_t(x_t) - \bar{g}_t(x_t) - g_t(x_{t+1}) + \bar{g}_t(x_{t+1}) - \frac{1}{4c}\mathbb{D}^2\left(\partial\bar{h}_t(x_t), \partial h_t(x_t)\right). \tag{16}$$

By Assumptions 4.1, and the definition of Wasserstein-1 distance, we have

$$g_t(x_{t+1}) - g_{t+1}(x_{t+1}) = \mathbb{E}_{\xi \sim P_{\xi,t}}[G(x_{t+1}, \xi)] - \mathbb{E}_{\xi \sim P_{\xi,t+1}}[G(x_{t+1}, \xi)] \geq -L_\xi W_1(P_{\xi,t+1}, P_{\xi,t}),$$

$$h_t(x_{t+1}) - h_{t+1}(x_{t+1}) = \mathbb{E}_{\xi \sim P_{\zeta,t}}[H(x_{t+1}, \xi)] - \mathbb{E}_{\xi \sim P_{\zeta,t+1}}[H(x_{t+1}, \xi)] \leq L_\zeta W_1(P_{\zeta,t+1}, P_{\zeta,t}).$$

Hence, we have

$$f_t(x_{t+1}) - f_{t+1}(x_{t+1}) \geq -L_\xi W_1(P_{\xi,t+1}, P_{\xi,t}) - L_\zeta W_1(P_{\zeta,t+1}, P_{\zeta,t}). \tag{17}$$

Taking expectation for both sides of (16), (17) under $\mathcal{F}_t$, we finish the proof after Corollaries 4.3 and 4.4. $\qquad\square$

## B.4. Proof of Theorem 4.8

*Proof.* Fix $c = \frac{\check{\mu}}{2}$. Taking the expectation with respect to $\mathcal{F}_0$ and summing over all $t$ in Lemma 4.5, we derive

$$\mathbb{E}\left[\sum_{t=0}^{n-1}\left(\mu_t + \frac{\rho_g + \rho_h}{2} - \frac{\check{\mu}}{2}\right)\|\bar{d}_t\|^2\Big|\mathcal{F}_0\right] \leq \mathbb{E}\left[f_0(x_0) - f_n(x_n) \mid \mathcal{F}_0\right] + \sum_{t=0}^{n-1}\left(\frac{C_g}{\mu_t N_{g,t}^{\alpha_g}} + \frac{C_h}{2\check{\mu}N_{h,t}^{\alpha_h}}\right)$$

$$+ \sum_{t=0}^{n-1}\left(L_\xi W_1\left(P_{\xi,t+1}, P_{\xi,t}\right) + L_\zeta W_1\left(P_{\zeta,t+1}, P_{\zeta,t}\right)\right). \tag{18}$$

Fix any $\epsilon > 0$, there exists $N > 0$ such that for any $n \geq N$, $W_1(P_{\xi,n}, P_\xi), W_1(P_{\zeta,n}, P_\zeta) \leq \epsilon$. Similar to (17), we have $|f_n(x_n) - f(x_n)| \leq W_1(P_{\xi,n}, P_\xi)L_\xi + W_1(P_{\zeta,n}, P_\zeta)L_\zeta$. Since $f(x) \geq \check{f}$, we have $f_n(x_n) \geq \epsilon(L_\xi + L_\zeta) + \check{f}$, which is lower bounded uniformly over $n \geq N$.

Combining Assumptions 4.6 and 4.7, the right hand side of (18) is upper bounded and the left hand side of (18) is greater than $\frac{\check{\mu}}{2}\mathbb{E}\left[\sum_{t=0}^{n-1}\|\bar{d}_t\|^2|\mathcal{F}_0\right]$, the first part of the theorem is proved by letting $n \to \infty$.

We proceed by contradiction for the second part of the theorem. Suppose there exists $\epsilon > 0$ and $a > 0$ such that

$$\mathbb{P}\left(\limsup_{t\to\infty}\|\bar{d}_t\| \geq \epsilon \mid \mathcal{F}_0\right) \geq a. \tag{19}$$

By Chebyshev's inequality, we have $\mathbb{P}\left(\|\bar{d}_t\| \geq \epsilon \mid \mathcal{F}_0\right) \leq \frac{\mathbb{E}[\|\bar{d}_t\|^2|\mathcal{F}_0]}{\epsilon^2}$. Since $\mathbb{E}\left[\|\bar{d}_t\|^2|\mathcal{F}_0\right]$ is finitely summable, there exists $T > 0$ such that $\sum_{t=T}^\infty \mathbb{P}\left(\|\bar{d}_t\| \geq \epsilon \mid \mathcal{F}_0\right) \leq \sum_{t=T}^\infty \frac{\mathbb{E}[\|\bar{d}_t\|^2|\mathcal{F}_0]}{\epsilon^2} < a$. Therefore,

$$\mathbb{P}\left(\limsup_{t\to\infty}\|\bar{d}_t\| \geq \epsilon \mid \mathcal{F}_0\right) = \mathbb{P}\left(\limsup_{t\to\infty: t\geq T}\|\bar{d}_t\| \geq \epsilon \mid \mathcal{F}_0\right) \leq \sum_{t=T}^\infty \mathbb{P}\left(\|\bar{d}_t\| \geq \epsilon \mid \mathcal{F}_0\right) < a. \tag{20}$$

This is a contradiction against (19). Hence, we could obtain that $\lim_{t\to\infty}\|\bar{d}_t\| = 0$ with probability 1. $\qquad\square$

## B.5. Proof of Lemma 4.9

*Proof.* We prove this by contradiction. Let $\Phi(x,\xi) = G(x,\xi) - G(\hat{x},\xi)$, $\phi_t(x) = g_t(x) - g_t(\hat{x})$, and $\bar{\phi}_t(x) = \bar{g}_t(x) - \bar{g}_t(\hat{x})$. Suppose there exist constants $\epsilon > 0$ and $a > 0$ such that $\mathbb{P}\left(\limsup_{t\to\infty}\left|\phi_t(x) - \bar{\phi}_t(x)\right| \geq \epsilon \mid \mathcal{F}_0\right) \geq a$.

By Chebyshev's inequality, we have $\mathbb{P}\left(\left|\phi_t(x) - \bar{\phi}_t(x)\right| \geq \epsilon \mid \mathcal{F}_0\right) \leq \frac{\mathbb{E}\left[\left|\phi_t(x) - \bar{\phi}_t(x)\right|^2|\mathcal{F}_0\right]}{\epsilon^2}$. Note that $\mathbb{E}_{\xi\sim P_{\xi,t}}[\Phi(x,\xi)] = \phi_t(x)$, and since $\Phi(x,\xi) \leq L_g R$, its variance is bounded by $L_g^2 R^2$. Thus, we have $\mathbb{E}\left[\left|\phi_t(x) - \bar{\phi}_t(x)\right|^2 \mid \mathcal{F}_t\right] \leq \frac{L_g^2 R^2}{N_{g,t}}$, and taking expectation with respect to $\mathcal{F}_0$, we get $\mathbb{E}\left[\left|\phi_t(x) - \bar{\phi}_t(x)\right|^2 \mid \mathcal{F}_0\right] \leq \frac{L_g^2 R^2}{N_{g,t}}$.

Summing over $t$, we obtain

$$\sum_{t=0}^\infty \mathbb{E}\left[\left|\phi_t(x) - \bar{\phi}_t(x)\right|^2 \mid \mathcal{F}_0\right] \leq \sum_{t=0}^\infty \frac{L_g^2 R^2}{N_{g,t}} < +\infty,$$

since $\sum_{t=0}^\infty N_{g,t}^{-\alpha_g} < +\infty$ holds for $\alpha_g < \frac{1}{2}$. Therefore, there exists $T > 0$ such that

$$\mathbb{P}\left(\limsup_{t\to\infty}\left|\phi_t(x) - \bar{\phi}_t(x)\right| \geq \epsilon \mid \mathcal{F}_0\right) = \mathbb{P}\left(\limsup_{t\to\infty, t\geq T}\left|\phi_t(x) - \bar{\phi}_t(x)\right| \geq \epsilon \mid \mathcal{F}_0\right) \leq \sum_{t=T}^\infty \mathbb{P}\left(\left|\phi_t(x) - \bar{\phi}_t(x)\right| \geq \epsilon \mid \mathcal{F}_0\right) < a.$$

This contradicts our assumption, and thus we conclude that $\lim_{t\to\infty}\left|\phi_t(x) - \bar{\phi}_t(x)\right| = 0$ with probability 1. The proof for the second part is similar, since the condition $\sum_{t=0}^\infty \frac{L_h^2 R^2}{N_{h,t}} < \infty$ also holds. $\qquad\square$

## B.6. Proof of Theorem 4.10

*Proof.* Let $x$ be an accumulation point of $\{x_t\}$. From the optimality condition (7), there exist $\bar{z}_{t+1} \in \partial\bar{g}_t(x_{t+1})$ and $\bar{v}_t \in \partial i_C(x_{t+1})$ for each $t$, such that

$$\bar{z}_{t+1} - \bar{y}_t + \mu_t\bar{d}_t + \bar{v}_t = 0. \tag{21}$$

We can assume $\lim_{t\to\infty} x_{n_t} = x$ where $x \in C$. Further, $\bar{y}_t$ and $\bar{z}_t$ are bounded due to the Lipschitz continuity of $G(x,\xi)$ and $H(x,\zeta)$. Thus, $\bar{v}_t$ is also bounded, and there exist accumulation points for $\{y_t\}$ and $\{z_t\}$. Without loss of generality, with the same subsequence, we assume $\bar{y}_{n_t} \to \bar{y}$ and $\bar{z}_{n_t+1} \to \bar{z}$. Therefore $\bar{y}_{n_t} \to \bar{y}$ and $\bar{z}_{n_t+1} \to \bar{z}$. By (21), we have

$$0 = \bar{z}_{n_t+1} - \bar{y}_{n_t} + \mu_{n_t}\bar{d}_{n_t} + \bar{v}_{n_t}. \tag{22}$$

By Theorem 4.8 and Assumption 4.7, $\lim_{t\to\infty} \mu_t\bar{d}_t = 0$ with probability 1. Thus, $\lim_{t\to\infty} \bar{v}_{n_t} = \bar{y} - \bar{z}$ with probability 1. Given that $\bar{v}_{n_t} \in \partial i_C(x_{n_t+1})$, the outer semicontinuity of $\partial i_C(\cdot)$ leads to $\lim_{t\to\infty} \bar{v}_{n_t} \in \partial i_C(x)$.

Finally, we prove that $\bar{y} \in \partial \bar{h}(\bar{x})$ and $\bar{z} \in \partial \bar{g}(\bar{x})$. As $\bar{y}_{n_t} \in \partial \bar{h}_t(x_{n_t})$, we have

$$\bar{y}_{n_t}^T (x - x_{n_t}) \leq \bar{h}_t(x) - \bar{h}_t(x_{n_t}), \text{ for all } x \in \mathbb{B}(\bar{x}, R). \tag{23}$$

Notice that

$$\left| \bar{h}_t(x_{n_t}) - h(\bar{x}) \right| \leq \left| \bar{h}_t(\bar{x}) - h(\bar{x}) \right| + L_h \left\| x_{n_t} - \bar{x} \right\|,$$

and for all $x \in \mathbb{B}(\bar{x}, R)$,

$$\left| \bar{h}_t(x) - h(x) \right| \leq \left| h_t(x) - \bar{h}_t(x) \right| + L_\zeta W_1(P_{\zeta,t}, P_\zeta).$$

Hence for all $x \in \mathbb{B}(\bar{x}, R)$, we have

$$\left| \bar{h}_t(x) - \bar{h}_t(x_{n_t}) - (h(x) - h(\bar{x})) \right| \leq L_h \left\| x_{n_t} - \bar{x} \right\| + L_\zeta W_1(P_{\zeta,t}, P_\zeta).$$

Combine with Lemma 4.9 and the fact that $x_{n_t} \to \bar{x}$ and $W_1(P_{\zeta,t}, P_\zeta) \to 0$, we conclude that $\bar{h}_t(x) - \bar{h}_t(x_{n_t}) \to h(x) - h(\bar{x})$ for all $x \in \mathbb{B}(\bar{x}, R)$ with probability 1. Hence, by letting $t \to \infty$ in (23), one obtains $\bar{y}^T(x - \bar{x}) \leq h(x) - h(\bar{x})$ for all $x \in \mathbb{B}(\bar{x}, R)$. This inequality yields $\bar{y} \in \partial h(\bar{x})$. The proof of $\bar{z} \in \partial \bar{g}(\bar{x})$ is analogous. Therefore, we prove that $\bar{x}$ is a DC critical point of $f$ with probability 1. $\square$

## C. An Adaptive Sampling Algorithm

### C.1. Proof of Theorem 5.3

*Proof.* Let $\mathcal{T}_1$ be the set of $t$ when Summable Condition is satisfied; $\mathcal{T}_2$ be the set of $t$ when Stepsize Norm Condition is satisfied. Supposed that when $t \geq T$, $t \in \mathcal{T}_1 \cup \mathcal{T}_2$. If $t \in \mathcal{T}_2$, we derive from Lemma 4.5 (the proof of Lemma 4.5 does not rely on Assumption 4.7.(b)) that

$$\mathbb{E}\left[f_t(x_t) - f_{t+1}(x_{t+1}) | \mathcal{F}_t\right] \geq (\mu_t + \frac{\rho_g + \rho_h}{2} - c_t)\mathbb{E}\left[\left\|\bar{d}_t\right\|^2 | \mathcal{F}_t\right]$$
$$- \left((\mu_{t-1} + \frac{\rho_g + \rho_h}{2}) - c_\mu - c_t\right)\left\|\bar{d}_{t-1}\right\|^2 - L_\xi W_1(P_{\xi,t+1}, P_{\xi,t}) - L_\zeta W_1(P_{\zeta,t+1}, P_{\zeta,t}).$$

Taking expectation under $\mathcal{F}_0$ from both side, we have

$$\mathbb{E}\left[f_t(x_t) - f_{t+1}(x_{t+1}) | \mathcal{F}_0\right] \geq (\mu_t + \frac{\rho_g + \rho_h}{2} - c_t)\mathbb{E}\left[\left\|\bar{d}_t\right\|^2 | \mathcal{F}_0\right] - \left((\mu_{t-1} + \frac{\rho_g + \rho_h}{2}) - c_\mu - c_t\right)\mathbb{E}\left[\left\|\bar{d}_{t-1}\right\|^2 | \mathcal{F}_0\right]$$
$$- L_\xi W_1(P_{\xi,t+1}, P_{\xi,t}) - L_\zeta W_1(P_{\zeta,t+1}, P_{\zeta,t}).$$

If $t \in \mathcal{T}_1 \cup \{0, 1, \cdots, T-1\}$, take expectation under $\mathcal{F}_0$, we have

$$\mathbb{E}\left[f_t(x_t) - f_{t+1}(x_{t+1}) | \mathcal{F}_0\right] \geq ((\mu_t + \frac{\rho_g + \rho_h}{2}) - c_t)\mathbb{E}\left[\left\|\bar{d}_t\right\|^2 | \mathcal{F}_0\right]$$
$$- \frac{C_g}{\mu_t N_{g,t}^{\alpha_g}} - \frac{C_h}{4c_t N_{h,t}^{\alpha_h}} - L_\xi W_1(P_{\xi,t+1}, P_{\xi,t}) - L_\zeta W_1(P_{\zeta,t+1}, P_{\zeta,t}).$$

Taking sum for each $t$, we have

$$\mathbb{E}\left[\sum_{t=0}^{n-1}\left(\mu_t + \frac{\rho_g + \rho_h}{2} - c_t\right)\left\|\bar{d}_t\right\|^2 | \mathcal{F}_0\right] \leq \mathbb{E}\left[f_0(x_0) - f_n(x_n) | \mathcal{F}_0\right] + \sum_{t=0}^{T-1}\left(\frac{C_g}{\mu_t N_{g,t}^{\alpha_g}} + \frac{C_h}{4c_t N_{h,t}^{\alpha_h}}\right)$$
$$+ \sum_{t=T, t\in\mathcal{T}_1}^{n-1}\left(\frac{C_g}{\mu_t N_{g,t}^{\alpha_g}} + \frac{C_h}{4c_t N_{h,t}^{\alpha_h}}\right) + \mathbb{E}\left[\sum_{t=T, t\in\mathcal{T}_2}^{n-1}\left((\mu_{t-1} + \frac{\rho_g + \rho_h}{2}) - c_{t-1} - c_\mu\right)\left\|\bar{d}_{t-1}\right\|^2 | \mathcal{F}_0\right] \tag{24}$$
$$+ \sum_{t=0}^{n-1}\left(L_\xi W_1(P_{\xi,t+1}, P_{\xi,t}) + L_\zeta W_1(P_{\zeta,t+1}, P_{\zeta,t})\right).$$

Hence, we could obtain that

$$
\mathbb{E}\left[\sum_{t=T-1}^{n-1} c_\mu \|\bar{d}_t\|^2 | \mathcal{F}_0\right] \leq \mathbb{E}\left[f_0\left(x_0\right) - f_n\left(x_n\right) | \mathcal{F}_0\right]
$$
$$
+ \sum_{t=0}^{T-1}\left(\frac{C_g}{\mu_t N_{g,t}^{\alpha_g}} + \frac{C_h}{4c_t N_{h,t}^{\alpha_h}}\right) + \sum_{t=T, t\in\mathcal{T}_1}^{n-1}\left(\frac{C_g}{\mu_t N_{g,t}^{\alpha_g}} + \frac{C_h}{4c_l N_{h,t}^{\alpha_h}}\right) + \sum_{t=0}^{n-1}\left(L_\xi W_1(P_{\xi,t+1}, P_{\xi,t}) + L_\zeta W_1(P_{\zeta,t+1}, P_{\zeta,t})\right).
$$
(25)

As we derived in Theorem 4.8, $f_n$ is lower bounded when $n$ is sufficiently large. Therefore, $\mathbb{E}\left[f_0\left(x_0\right) - f_n\left(x_n\right) | \mathcal{F}_0\right]$ is bounded above. By the definition of Summable Condition and Assumption 4.7, we have

$$
\sum_{t=T, t\in\mathcal{T}_1}^{n-1}\left(\frac{C_g}{\mu_t N_{g,t}^{\alpha_g}} + \frac{C_h}{4c_l N_{h,t}^{\alpha_h}}\right) \leq \sum_{t=T, t\in\mathcal{T}_1}^{n-1}\left(\frac{C_g}{\check{\mu}\hat{N}_{g,t}^{\alpha_g}} + \frac{C_h}{4c_l \hat{N}_{h,t}^{\alpha_h}}\right).
$$

Moreover, as we derived in Corollary 4.3, $\|\bar{d}_t\| \leq \frac{2(L_g+L_h)}{\mu_t} \leq \frac{2(L_g+L_h)}{\check{\mu}}$, which is bounded. It follows that $\mathbb{E}\left[\sum_{t=0}^{T-2}\|\bar{d}_t\|^2 | \mathcal{F}_0\right] < \infty$. Take $n \to \infty$, since the right-hand side of (25) is bounded, we have

$$
\lim_{t\to\infty} \mathbb{E}\left[\sum_{t\geq 0} \|\bar{d}_t\|^2 | \mathcal{F}_0\right] < \infty.
$$
(26)

Supposed that when $t \geq T$, $t \in \mathcal{T}_1 \cup \mathcal{T}_2$. If $t \in \mathcal{T}_1$, we have

$$
N_{g,t} \geq \hat{N}_{g,t}, \text{ and } N_{h,t} \geq \hat{N}_{h,t}, \text{ where } \sum_{t\geq 0}\left(\hat{N}_{h,t}^{-\alpha_h} + \hat{N}_{g,t}^{-\alpha_g}\right) < \infty.
$$
(27)

If $t \in \mathcal{T}_2$, we have

$$
\begin{aligned}
\frac{C_g}{\hat{\mu} N_{g,t}^{\alpha_g}} + \frac{C_h}{4(\hat{\mu} + \frac{\rho_g+\rho_h}{2} - c_\mu)N_{h,t}^{\alpha_h}} &\leq \frac{C_g}{\mu_t N_{g,t}^{\alpha_g}} + \frac{C_h}{4(\mu_t + \frac{\rho_g+\rho_h}{2} - c_\mu)N_{h,t}^{\alpha_h}} \\
&\leq \frac{C_g}{\mu_t N_{g,t}^{\alpha_g}} + \frac{C_h}{4c_t N_{h,t}^{\alpha_h}} \\
&\leq \left(\mu_{t-1} + \frac{\rho_g+\rho_h}{2} - c_\mu - c_{t-1}\right)\|\bar{d}_{t-1}\|^2 \\
&\leq \left(\hat{\mu} + \frac{\rho_g+\rho_h}{2} - c_\mu - c_{t-1}\right)\|\bar{d}_{t-1}\|^2,
\end{aligned}
$$
(28)

where the first and last inequality is due to Assumption 4.7. The second and third inequality is due to the definition of Stepsize Norm Condition (11). (28) implies that

$$
\frac{C_g}{\hat{\mu} N_{g,t}^{\alpha_g}} + \frac{C_h}{4(\hat{\mu} + \frac{\rho_g+\rho_h}{2})N_{h,t}^{\alpha_h}} \leq \left(\hat{\mu} + \frac{\rho_g+\rho_h}{2}\right)\|\bar{d}_{t-1}\|^2.
$$
(29)

By (27), (29) and (26), it follows that

$$
\begin{aligned}
\sum_{t\geq 0}\left(\frac{C_g}{\hat{\mu} N_{g,t}^{\alpha_g}} + \frac{C_h}{4(\hat{\mu} + \frac{\rho_g+\rho_h}{2})N_{h,t}^{\alpha_h}}\right) &\leq \sum_{t=0}^{T-1}\left(\frac{C_g}{\hat{\mu} N_{g,t}^{\alpha_g}} + \frac{C_h}{4(\hat{\mu} + \frac{\rho_g+\rho_h}{2})N_{h,t}^{\alpha_h}}\right) \\
&+ \sum_{t\geq T, t\in\mathcal{T}_1}\left(\frac{C_g}{\hat{\mu}\hat{N}_{g,t}^{\alpha_g}} + \frac{C_h}{4(\hat{\mu} + \frac{\rho_g+\rho_h}{2})\hat{N}_{h,t}^{\alpha_h}}\right) + \sum_{t\geq T, t\in\mathcal{T}_2}\left(\hat{\mu} + \frac{\rho_g+\rho_h}{2}\right)\|\bar{d}_{t-1}\|^2 < \infty.
\end{aligned}
$$

Hence, we derive from the above that 4.7 (b) holds. $\square$

## C.2. Adaptive ospDCA: A Full Version

---

**Algorithm 3** Adaptive ospDCA

---

**Require:** Initial point $x_0$, initial parameter $\mu_0$, $c_0$, $N_{g,0}$ and $N_{h,0}$, error estimation parameter $\alpha_g \in (0, 1/2)$, $\alpha_h \in (0, 1)$ with corresponding $C_g, C_h$ defined in Corollaries 4.3 and 4.4, stepsize norm condition parameter $c_\mu, c_l > 0$, sample size upper bound sequence $\{\hat{N}_{g,t}\}$ and $\{\hat{N}_{h,t}\}$ which satisfy $\sum_{t \geq 0} \left( \hat{N}_{h,t}^{-\alpha_h} + \hat{N}_{g,t}^{-\alpha_g} \right) < \infty$, stepsize upper bound $\hat{\mu}$ and lower bound $\check{\mu}$ that satisfy $\hat{\mu} > \check{\mu}$.

1: **for** $t = 0, 1, 2, \cdots$ **do**
2:     Generate iid samples $\{\xi^{t,i}\}_{i=1}^{N_{g,t}}$ and $\{\zeta^{t,i}\}_{i=1}^{N_{h,t}}$ from the distribution of $\xi$ and $\zeta$, which are independent of the past.
3:     Set $\bar{g}_t(x) = \frac{1}{N_{g,t}} \sum_{i=1}^{N_{g,t}} G\left(x, \xi^{t,i}\right)$, $\bar{h}_t(x) = \frac{1}{N_{h,t}} \sum_{i=1}^{N_{h,t}} H\left(x, \zeta^{t,i}\right)$, and select $\bar{y}_t \in \partial \bar{h}_t(x_t)$.
4:     Solve the convex subproblem to obtain $\bar{d}_t$:

$$\underset{d}{\text{minimize}} \quad \bar{g}_t(x_t + d) - \bar{h}_t(x_t) - \bar{y}_t^T d + \frac{1}{2}\mu_t \|d\|^2$$
$$\text{subject to} \quad x_t + d \in C.$$

5:     Take the step $x_{t+1} = x_t + \bar{d}_t$.
6:     Update $\check{\mu} \leq \mu_{t+1} \leq \hat{\mu}$, $c_{t+1}, N_{g,t+1}$ and $N_{h,t+1}$ such that one of the followings stands:

    1. $\left(\mu_t + \frac{\rho_g + \rho_h}{2} - c_\mu - c_t\right) \|\bar{d}_t\|^2 \geq \frac{C_g}{\mu_{t+1} N_{g,t+1}^{\alpha_g}} + \frac{C_h}{4c_t N_{h,t+1}^{\alpha_h}}$, and $c_{t+1} \leq \mu_{t+1} + \frac{\rho_g + \rho_h}{2} - c_\mu$,

    2. $N_{g,t+1} \geq \hat{N}_{g,t+1}$, $N_{h,t} \geq \hat{N}_{h,t+1}$, and $c_l \leq c_{t+1} \leq \mu_{t+1} + \frac{\rho_g + \rho_h}{2} - c_\mu$.

7: **end for**

---

# D. More Experimental Results

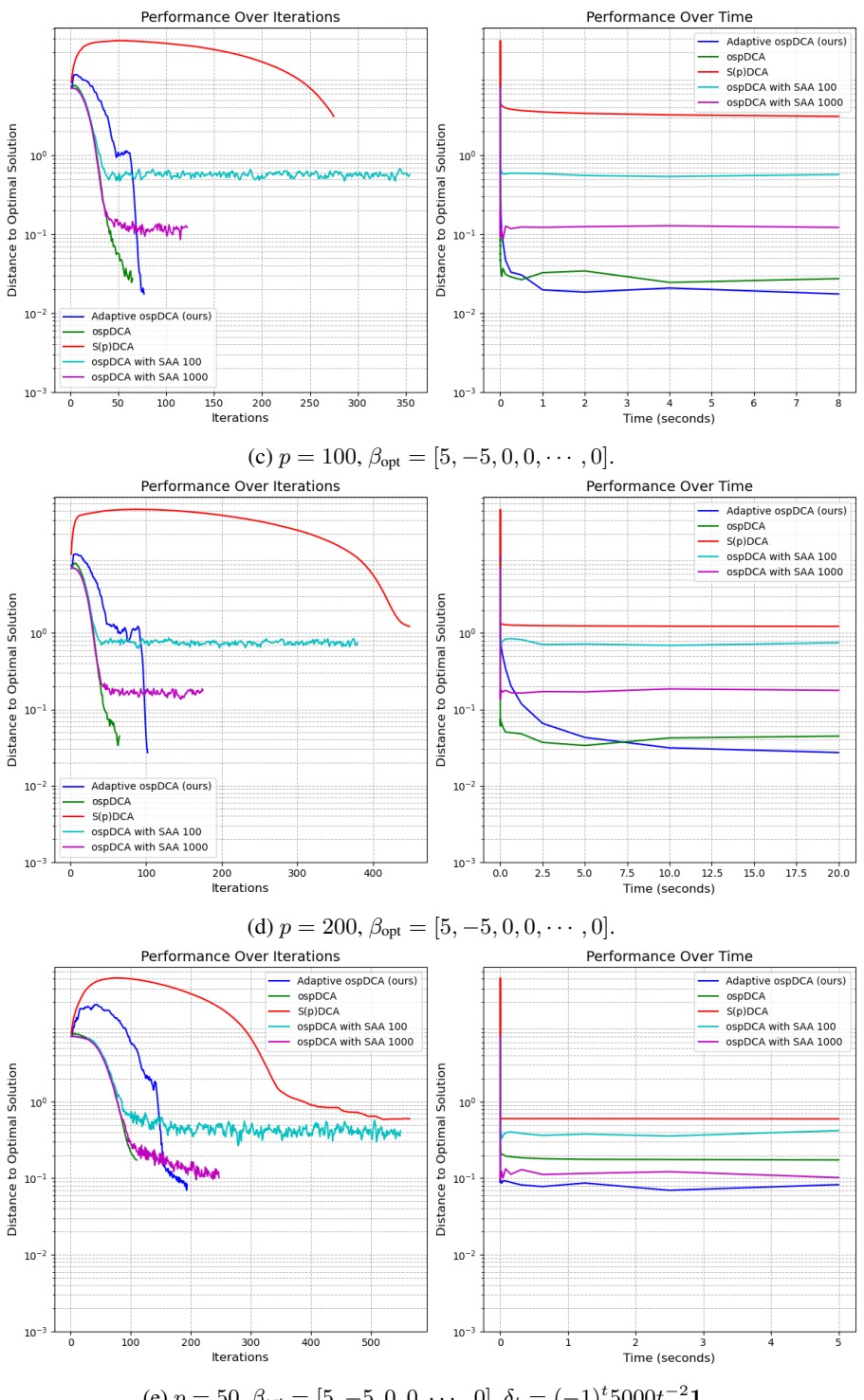

(c) $p = 100$, $\beta_{\text{opt}} = [5, -5, 0, 0, \cdots, 0]$.

(d) $p = 200$, $\beta_{\text{opt}} = [5, -5, 0, 0, \cdots, 0]$.

(e) $p = 50$, $\beta_{\text{opt}} = [5, -5, 0, 0, \cdots, 0]$, $\delta_t = (-1)^t 5000 t^{-2} \mathbf{1}_p$.

*Figure 3.* Algorithm behavior for online sparse robust regression.

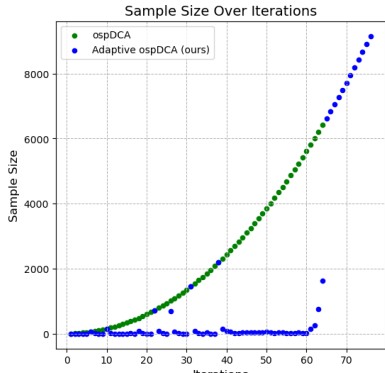

Sample size for experiment (c).

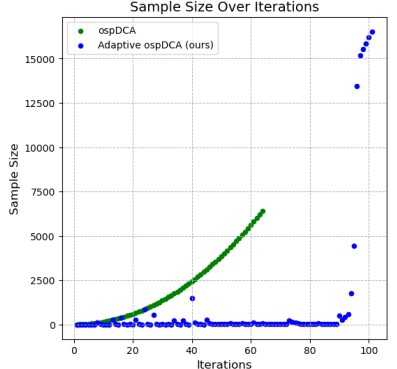

Sample size for experiment (d).

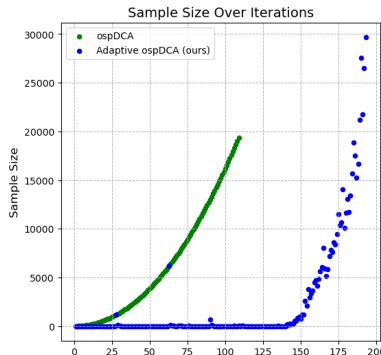

Sample size for experiment (e).

*Figure 4.* Sample size per iteration.

