# OpenReview forum: "An Online Adaptive Sampling Algorithm for Stochastic Difference-of-convex Optimization with Time-varying Distributions"
_ICML.cc/2025/Conference — ICML 2025 oral_

### Official Review · Reviewer_Fi1S · 2025-03-02

**Overall Recommendation:** 4

**Summary:**

In this paper, the authors propose an online adaptive sampling algorithm for solving nonsmooth DC problems under time-varying distributions.
Their major technique is the development of a convergence rate for the sample average approximation of subdifferential mapping.
Based on the technique, they show their algorithm converges subsequentially to DC critical points almost surely under proper assumptions.

## update after rebuttal
The author's reply makes sense, so I keep my score.

**Claims And Evidence:**

The authors propose stochastic algorithms to solve the nonsmooth DC problem, and they provide the asymptotic convergence of their algorithm under mild assumptions.
Their theoretical findings are both solid and interesting.
My only concern for this paper is what machine learning applications belong to the class of stochastic DC problems under time-varying distributions.
I have noticed the author listed the online sparse robust regression as an application of their problem.
Could you provide more machine-learning examples of the nonsmooth DC problem?
In addition, is there any real-world time-varying data distribution that satisfies Assumption 4.6, could you provide some examples?

**Essential References Not Discussed:**

I think most related works are cited and discussed.

**Experimental Designs Or Analyses:**

The experimental designs are valid and sound.

**Methods And Evaluation Criteria:**

The proposed methods and evaluation criteria for solving the problem make sense to me, and the author provides a rigorous theoretical analysis of their methods.

**Other Comments Or Suggestions:**

I found some small typos in the paper:

* In line 209, did you miss a square for the $\delta_n$?
* In assumption 4.1, I think it is better to say $\rho_g$-strongly convex.
* In Lemma 4.2, there is no definition of the function $f_t$, although it is not hard to guess.
* The conclusion and future work section is missing in this paper.
* In Line 708, what is $\alpha_h$? Should it be $\alpha'$?

**Other Strengths And Weaknesses:**

Since there are many works developing stochastic methods with non-asymptotic convergence rates for nonconvex nonsmooth optimization problems.
Is it possible to show the proposed methods with a non-asymptotic convergence rate for solving the stochastic DC problems?

**Questions For Authors:**

See above.

**Relation To Broader Scientific Literature:**

See above.

**Theoretical Claims:**

See above.

---

> ### Author Rebuttal · Authors · 2025-03-27
>
> Thank you for your careful reading and valuable feedback. We address your comments as follows:
>
> 1. There are some other machine learning problems with a nonsmooth DC structure. It is well known that piece-wise linear functions are DC. In order to guarantee both robustness and continuity, they could serve as the surrogate loss functions for binary classification. A direct example is calculating AUROC (area under ROC curve) of a predictive function $h(\cdot,w)$:
>
> $$AUROC(h(\cdot,w)) = Pr(h(x_+,w) > h(x_-,w)) = E_{x_+ \sim P_+, x_- \sim P_-} [\mathbb{1}(h(x_+,w) > h(x_-,w))],$$
>
> where $P_+$ is the distribution of positive examples, $P_-$ is the distribution of negative examples, and $h(x,w) : \mathcal{X} \to \mathbb{R}$ is a predictive function parameterized by a vector $w \in \mathbb{R}^d$. $\mathbb{1}(\cdot)$ is an indicator function of a predicate.
>
> Let $\ell(w; x, x') = \ell(h(x',w) - h(x,w))$ denote a pairwise surrogate loss for a positive-negative pair $(x, x')$ to approximate $\mathbb{1}(\cdot)$. If $h(x,\cdot)$ and the surrogate loss $\ell(\cdot~; x, x')$ are both piecewise linear, the minimization problem with regard to $w$ can be formulated as a stochastic DC problem.
>
> For the time-varying data distribution, section 6 provides an example of time-varying multi-variable normal distributions that satisfy the assumption. Since the Wasserstein-1 distance of some common distributions is easy to calculate or control, it is not hard to construct examples of time-varying exponential or uniform distributions that satisfy the assumption. We will add some examples in the final version of the paper. A simple but direct real-world example is problems with finite outliers or finite times of distribution shifts (due to the change of environment).
>
> 2. Establishing non-asymptotic rates remains an interesting but challenging problem. The derivation of Theorems 3.5.(ii) and 3.6.(iii) in [1] suggests that proving strict non-asymptotic rates still requires a smoothness assumption on either $g$ or $h$, even in the deterministic setting. Without such assumptions, obtaining non-asymptotic guarantees becomes significantly more difficult. This challenge persists unless a relaxed convergence criterion, such as nearly $\epsilon$-critical points, is considered.
>
> [1] Hoai An Le Thi, Van Ngai Huynh, Tao Pham Dinh, and Hoang Phuc Hau Luu. Stochastic difference-of-convex-functions algorithms for nonconvex programming. SIAM Journal on Optimization, 32(3):2263–2293, 2022.
>
> 3. We sincerely appreciate your meticulous attention to detail. We agree that saying "$\rho$-strongly complex" is better. For the typos in lines 209 and 708, we will carefully revise them and ensure accuracy in the final version. We will also add a conclusion section. For reference, the definitions of $f_t$, $g_t$, and $h_t$ can be found in line 243 (left).
>
> Thank you again for your valuable feedback. Your insights are greatly appreciated and will help improve the clarity and rigor of our work.

---

### Official Review · Reviewer_AmGZ · 2025-03-10

**Overall Recommendation:** 4

**Summary:**

The authors address the minimization of a function defined as the difference of two convex functions.
Moreover, these two convex functions are expressed as the expectations of random functions.
The authors then propose online estimators based on an adaptive sampling algorithm.

**Claims And Evidence:**

The proofs seem solid.

**Essential References Not Discussed:**

Not to my knowledge.

**Experimental Designs Or Analyses:**

N/A

**Methods And Evaluation Criteria:**

The simulation work is somewhat limited but very promising. However, the methods have not been applied to real data.

**Other Comments Or Suggestions:**

It's just a suggestion, but wouldn't the proofs have been simpler (at least to establish the almost sure convergence of the estimators) by using Robbins-Siegmund's theorem?

**Other Strengths And Weaknesses:**

The proposed methods and results are very interesting.
However, I have a few remarks:

- I believe the discussion on the number of data points to generate should be expanded. The approach appears computationally intensive since, at each step, a larger dataset needs to be simulated than in the previous step. That said, I do agree that the simulations suggest the proposed method is computationally faster than existing methods.
- The paper is somewhat difficult to read, partly due to the complexity of the problem studied and partly because of the large number of technical lemmas in the core of the paper (e.g., Lemmas 3.3 and 4.2). These lemmas do not seem to aid comprehension and instead make the paper heavier. It would be better to move them to the appendix, freeing up space for more in-depth simulations or an application to real data.
- The proofs are challenging to follow, with some steps moving quite quickly.

**Questions For Authors:**

N/A

**Relation To Broader Scientific Literature:**

The key contribution consists in considering subdifferentiable sets while in the existing literature, authors often consider smooth functions.

**Theoretical Claims:**

The proofs seem true.

---

> ### Author Rebuttal · Authors · 2025-03-28
>
> Thank you for your careful reading and valuable feedback. We address your concerns as follows:
>
> 1. We appreciate your comments on the simulation work. Our primary goal was to verify the theoretical validity of our method rather than to apply it to real data. The numerical result has demonstrated that our algorithm is efficient and effective in a simple but common problem. Your suggestion is very meaningful, and we plan to apply our algorithm to real and large-scale data in future work.
>
> 2. Regarding the sublinearly increasing dataset size, this aspect is inherent to our approach and difficult to avoid, as it ensures the necessary accuracy of the algorithm at each step. The choice of sample size and step size remains a fascinating topic in stochastic optimization. Even in standard smooth SGD without further assumptions, a non-vanishing step size requires a sublinearly increasing sampling size to guarantee convergence, since the variance reduction procedure is unavoidable.
>
>    In our paper, the proximal terms $\mu_t$ for each DC subproblem can be pre-selected arbitrarily, as long as they are upper and lower bounded by positive constants. At time $t$, our $O(t^{2+\epsilon})$ sample size for $g$ and $O(t^{1+\epsilon})$ sample size for $h$ match the order of the smooth case. Moreover, our adaptive strategy is also designed to control the sample sizes when the current iterate is far from the critical points. Our sample size has already been an almost tight result, thanks to the novel $O(\sqrt{p/n})$ pointwise convergence rate for the SAA of subdifferential mappings.
>
> 4. To improve readability, we will consider moving some of the technical lemmas to the appendix while ensuring that the main ideas remain accessible in the main text. We also recognize that certain proof steps move quickly, and we will provide additional explanations and guidance to enhance clarity. We are also ready to include an additional simulation in our paper.
>
> 5. We agree that Robbins-Siegmund's theorem could be used in Theorems 4.8 and 5.3. However, the major part of our proof could not be replaced by this. The convergence of some series, such as the ones on the right-hand side of (18), remains essential; Robbins-Siegmund's theorem cannot simplify these. Regarding Theorem 5.3, we must separately consider iteration steps that satisfy the Summable Condition and the Stepsize Norm Condition, even if the theorem is used.
>
>     That being said, we acknowledge Robbins-Siegmund's theorem as an insightful tool for guiding our convergence analysis, and we will add a remark discussing its relevance. However, the proofs themselves would not be significantly simplified by directly applying the theorem.
>
> Thank you again for your thoughtful feedback. We will incorporate these improvements to enhance the clarity and accessibility of our work.

---

### Official Review · Reviewer_rStE · 2025-03-13

**Overall Recommendation:** 4

**Summary:**

The paper studies stochastic difference-of-convex (DC) optimization. The analysis accounts for distribution shifts, and for non-smoothness of the components is derived, introducing some non-trivial technical contributions. The obtained algorithm is validated in a numerical experiment.

**Claims And Evidence:**

The paper is extremely well written in my opinion. All claims are motivated, situated with respect to prior work, which makes is rather easy to follow even for non-experts.

**Essential References Not Discussed:**

I am not aware of essential references which are missed.

**Experimental Designs Or Analyses:**

I did not check the validity of the experiment itself.

**Methods And Evaluation Criteria:**

Then proposed methods make sense, and are essentially generalizations of prior discussed work.

**Other Comments Or Suggestions:**

Additional comments:

- Remark 2 is unclear to me, what does an isomorphic map imply here and why?

**Other Strengths And Weaknesses:**

As I mentioned, the paper is extremely well written in my opinion.

The technical contributions, even beyond the end results, are quite nice.

In particular, accounting for distribution shifts in the descent lemma by incorporating a Wasserstein distance term (Lemma 4.5) is a very interesting idea, which I have not seen before. This approach is really nice, and can be applied to many other optimization scenarios.

**Questions For Authors:**

- Assumption 3.2 confuses me - wouldn't this correspond later to the gradient being Lipschitz? If so, this would require smoothness, which the authors are trying to avoid. I kindly ask the authors clarify this issue.

Questions to the authors:

- Is the idea of incorporating a Wasserstein distance term in the descent lemma (Lemma 4.5) novel in this work?

- It would be nice to derive non-asymptotic rates, which seems doable via the provided descent lemma (perhaps under further quantifiable assumptions). Is there a clear difficulty in doing so? It is fine to leave this for future work nonetheless.

**Relation To Broader Scientific Literature:**

The authors do a fantastic job in my opinion situating this work with respect to prior works on this topic. Even small technical derivations are compared to prior analogous results, as well as the main ideas.

**Theoretical Claims:**

I did not closely check the correctness of the claims. I believe they are overall correct, as the proofs are sketched and motivated in the main text in a rather convincing manner, and the main claim generalize prior known results.

---

> ### Author Rebuttal · Authors · 2025-03-27
>
> Thank you for your careful reading and valuable feedback. We address your questions as follows:
>
> 1. **Remark 2:** The main idea is that if there exists an isomorphic mapping between the probability spaces of the random variables $\xi$ and $\zeta$ associated with $G$ and $H$ (e.g., if $\xi$ and $\zeta$ originate from the same probability space, share the same distribution, or even are the same random variable), then we only need to sample the variable with the higher sampling demand. The other variable’s samples can be obtained directly via this mapping, reducing the overall sampling cost without affecting convergence.
>
> 2. **Assumption 3.2** concerns only the Lipschitz continuity of the original function, not its gradient. The function itself may not even be differentiable, e.g., $\varphi(x,\omega)=|x-\omega|$ with corresponding $L_{\varphi}=1$. We require $\varphi(\,\cdot,\omega)$ to be Lipschitz continuous in $x$ with a universal constant $L_{\varphi}$, independent of $\omega$. Our result does not assume any smoothness.
>
> 3. The idea of using the Wasserstein distance to detect distribution shifts in online or decision-dependent optimization is not new, see [1,2]. However, we are the first to directly incorporate a Wasserstein distance term into the descent lemma and establish almost sure convergence.
>
> [1] Drusvyatskiy D, Xiao L. Stochastic optimization with decision-dependent distributions. Mathematics of Operations Research, 2023, 48(2): 954-998.
>
> [2] Che E, Dong J, Tong X T. Stochastic gradient descent with adaptive data. arXiv preprint arXiv:2410.01195, 2024.
>
> 4. Establishing non-asymptotic rates remains an interesting but challenging problem. The derivation of Theorems 3.5.(ii) and 3.6.(iii) in [3] suggests that proving strict non-asymptotic rates still requires a smoothness assumption on either $g$ or $h$, even in the deterministic setting. Without such assumptions, obtaining non-asymptotic guarantees becomes significantly more difficult. This challenge persists unless a relaxed convergence criterion, such as nearly $\epsilon$-critical points, is considered.
>
> [3] Hoai An Le Thi, Van Ngai Huynh, Tao Pham Dinh, and Hoang Phuc Hau Luu. Stochastic difference-of-convex-functions algorithms for nonconvex programming. SIAM Journal on Optimization, 32(3):2263–2293, 2022.
>
> Thank you again for your valuable feedback. We appreciate your insights and will carefully consider them in our revisions.

---

### Official Review · Reviewer_q3uU · 2025-03-18

**Overall Recommendation:** 4

**Summary:**

This paper proposes algorithms for solving a stochastic DC program in a time-varying setting. Specifically, the distributions used to define stochastic convex components may vary over time and are assumed to converge to the true distributions. The proposed algorithm is a variant of the classic DC algorithm and uses SAA to estimate the current distributions on the fly. The main result is an almost sure convergence guarantee to DC critical points, up to taking subsequences. To prove this result, the authors develop a new upper bound on the estimation error of the SAA scheme for convex subdifferentials in terms of the excess of one set over another. Overall, I think the contribution is interesting and meaningful for solving a difficult stochastic DC program.

**Claims And Evidence:**

Yes.

**Essential References Not Discussed:**

No.

**Experimental Designs Or Analyses:**

No.

**Methods And Evaluation Criteria:**

Yes.

**Other Comments Or Suggestions:**

See above.

**Other Strengths And Weaknesses:**

Overall, I think this paper makes good and interesting contributions to DC programming in the modern stochastic setting. I only have the following comments.

- L019, right: I suggest using the term "DC critical point" here rather than "critical point," since the latter is sometimes used interchangeably with "stationary point" and has a totally different meaning compared to a DC critical point.

- L127, right: It seems $\mathbb{P}(\Omega)$ is a set of distributions. Hence, it is not clear to me what the meaning of $\xi \sim \mathbb{P}(\Omega)$ is.

- L160, left: It seems that $g, h$ here are convex extended-real-valued functions, since you need to use the indicator function $i_C:\mathbb{R}^p\to\overline{\mathbb{R}}$ to represent the constraint $x \in C$. However, I cannot find their concrete definitions.

- L160, right: The function $\phi$ should be $\varphi$?

- L280, right: Compared with the convergence conditions in L328, left, these two summable conditions seem a bit stringent. It would be illustrative if a concrete example were discussed that satisfies this summable condition.

- L806: Lemma 4.3 should be Corollary 4.3.

- L869: \bar{z}_{n}  should be \bar{z}_{n_t + 1}.

- Some papers are listed in the references without a citation in the main paper, e.g., (Kantorovich, 1958), (Goldstein, 1977), (Geyer, 1994), (Mehta, 2016, 2014), (Liu et al., 2018) and many others.

**Questions For Authors:**

See above.

**Relation To Broader Scientific Literature:**

The new algorithmic framework in this paper is built on the classic DCA and SAA schemes. The authors propose new theoretical results to determine the number of samples sufficient for almost sure convergence to a DC critical point.

**Theoretical Claims:**

Yes, I have checked some of the proofs in Appendices A and B, and they look good to me.

---

> ### Author Rebuttal · Authors · 2025-03-27
>
> Thank you for your careful reading and valuable feedback. Below, we provide clarifications regarding the comments you raised:
>
> 1. **L019, right:** Thank you for pointing this out. In this paper, "critical point" specifically refers to a "DC critical point," following the convention in other DC programming literature. We will add a paragraph in Section 2 to clarify this distinction.
>
> 2. **L127, right:** Thanks for pointing out this improper statement. We will change the statement to the following:
>    *Let $\mathbb{P}(\Omega)$ denote the set of Radon probability measures on $\Omega$, where each measure $P \in \mathbb{P}(\Omega)$ has a finite first moment. That is, $\mathbb{E}_{\xi \sim P}[d(\xi, \xi_0)] < \infty$ for some $\xi_0 \in \Omega$.*
>
> 3. **L160, left:** We will modify the definition of $g$ to be extended-real-valued functions so that it covers the indicator function. The function $h$ is real-valued to avoid making $f-g = -\infty$. Thank you again for pointing this out.
>
> 4. **L280, right:** These conditions seem to be necessary. In Section 6, we provide an example of online sparse robust regression to illustrate their role. We will add another example directly after the summable assumptions for further illustrations, as suggested by the reviewer.
>
> 5. **L160, right; L806; L869; references:** We appreciate your careful attention to detail. We will correct these typographical errors and remove uncited references accordingly.
>
> Thank you again for your valuable feedback. We will carefully incorporate these revisions to improve the clarity and precision of our work.

---

### Decision · Program_Chairs · 2025-05-01

**Decision:**

Accept (oral)

**Comment:**

This paper investigates stochastic optimization of difference-of-convex (DC) functions, where both components are expressed as expectations over random functions and the data distribution may shift over time. The authors develop an online adaptive sampling algorithm and provide rigorous asymptotic convergence guarantees to DC-critical points under mild assumptions. A key technical innovation is a convergence rate for the sample average approximation of subdifferential mappings, with the analysis carefully incorporating distributional shifts via Wasserstein distances. The main convergence result  is nontrivial and appears technically sound.